# Native Active Perception as Reasoning for Omni-Modal Understanding

**Zhenghao Xing** [1 * †]  **Ruiyang Xu** [2 * †]  **Yuxuan Wang** [3 *]  **Jinzheng He** [3]  **Ziyang Ma** [2 4 †]  **Qize Yang** [3]
**Yunfei Chu** [3]  **Jin Xu** [3]  **Junyang Lin** [3]  **Chi-Wing Fu** [1]  **Pheng-Ann Heng** [1]

## Abstract

Passive models for long video understanding typically rely on a "watch-it-all" paradigm, processing frames uniformly regardless of query difficulty, causing computational cost to grow with video duration. Although interactive frameworks have emerged, they often rely on global pre-scanning, and their context cost still scales with video length. We propose **OmniAgent**, the first native omni-modal agent that formulates video understanding as a POMDP-based iterative **Observation-Thought-Action** cycle. OmniAgent executes on-demand actions to selectively distill audio-visual cues into a persistent textual memory, effectively decoupling reasoning complexity from raw video duration. To operationalize this, we introduce (1) **Agentic Supervised Fine-Tuning** to bootstrap native active perception via best-of-N trajectory synthesis with dual-stage quality control, and (2) **Agentic Reinforcement Learning** with **TAURA** (Turn-aware Adaptive Uncertainty Rescaled Advantage), which leverages turn-level entropy to steer credit assignment toward pivotal discovery turns. Crucially, OmniAgent exhibits positive test-time scaling, where performance improves as the number of reasoning turns increases, validating the efficacy of active perception. Empirical results across ten benchmarks (e.g., VideoMME, LVBench) demonstrate that OmniAgent achieves state-of-the-art performance among open-source models. Notably, on LVBench, our 7B agent outperforms the $10\times$ larger Qwen2.5-VL-72B (50.5% vs. 47.3%). We release our code and model at https://github.com/HarryHsing/OmniAgent.

---

[*]Equal contribution [†]Work done during an internship at Qwen.
[1]The Chinese University of Hong Kong [2]Shanghai Jiao Tong University [3]Qwen Team, Alibaba Group [4]Nanyang Technological University. Correspondence to: Jin Xu <jxu3425@gmail.com>, Pheng-Ann Heng <pheng@cse.cuhk.edu.hk>.

*Proceedings of the $43^{rd}$ International Conference on Machine Learning*, Seoul, South Korea. PMLR 306, 2026. Copyright 2026 by the author(s).

## 1. Introduction

Recent advances in scaling laws have propelled large language models (LLMs) toward general-purpose intelligence, extending capabilities into the visual (Li et al., 2022; Liu et al., 2023; Bai et al., 2025; Lin et al., 2024) and auditory domains (Chu et al., 2024; Xu et al., 2025a). Despite these strides, current paradigms largely treat multimodal perception as the processing of static snapshots or fixed-window streams. This approach clashes with the nature of human perception, which functions as an active, continuous interrogation of intertwined signals. More critically, the high dimensionality of spatiotemporal data imposes a prohibitive constraint: computational cost scales super-linearly with sequence length. This renders passive end-to-end processing computationally intractable for long-form video understanding, creating a central bottleneck in open-world multimodal modeling.

To mitigate this computational burden, prior work has explored agentic adaptations. One branch utilizes LLMs as controllers to invoke modality-specialized tools (Fan et al., 2024; Zhang et al., 2025b; Long et al., 2026). While expedient, this reliance on intermediate modules creates an information bottleneck, severing the gradient flow between reasoning and perception. A second branch pursues "thinking with images" by integrating transformation tools, such as temporal clipping (Zhang et al., 2026; Yang et al., 2026) or spatial zooming (Shen et al., 2025a), directly within the multimodal large language models (MLLMs)'s chain-of-thought. However, these methods often retain a *semi-passive* nature: they typically require a global pre-scan of the video or maintain a dense visual buffer to decide "where to look," failing to truly decouple reasoning complexity from video duration. Consequently, they struggle to scale to hour-long videos where raw pixel retention is infeasible.

In response, we propose **OmniAgent** (Figure 1), a framework that reimagines MLLMs not as passive observers, but as native active perceivers. We formulate audio-visual exploration as a Partially Observable Markov Decision Process (POMDP), where the agent performs a strict *information distillation*. Through an iterative **Observation-Thought-Action (OTA)** cycle, the agent actively browses (frames), listens (audio), and watches (audio-visual clips), distilling

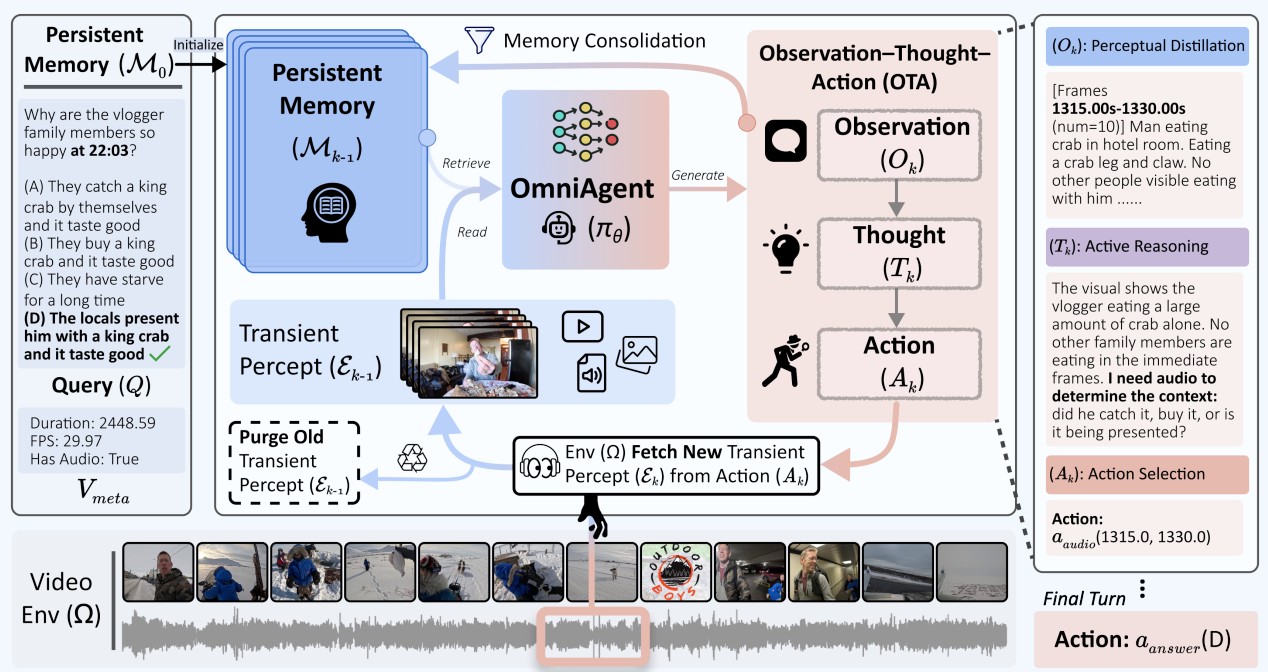

*Figure 1.* **The OmniAgent Framework for Native Active Perception.** Unlike passive methods that process video frames uniformly, OmniAgent treats perception as an iterative reasoning process via an *Observation-Thought-Action (OTA)* cycle. Conditioned on a specific query, the agent executes on-demand actions to selectively gather audio-visual cues, distilling high-dimensional transient percepts into a persistent textual memory until sufficient evidence is gathered to produce the final answer.

high-dimensional transient percepts into a *persistent textual memory*. This architecture ensures that the agent's internal state depends solely on the complexity of the reasoning trace rather than the raw duration of the video. This formulation gives rise to an emergent *test-time scaling* property: the model adaptively allocates more computational steps to resolve harder queries, analogous to System-2 reasoning, where inference-time compute is dynamically expended to resolve ambiguity. Unlike prior agentic approaches that delegate multimodal perception to external modules, Omni-Agent is a single native omni model: the environment only performs raw media extraction (returning frames, audio segments, or video clips), and all perception and reasoning are performed by the same model that acts.

To operationalize this, we introduce a two-stage optimization. First, we bootstrap native active perception via **Agentic Supervised Fine-Tuning (SFT)**, a best-of-N trajectory synthesis pipeline with dual-stage quality control (Sec. 3.2). Second, we refine the policy via **Agentic Reinforcement Learning (RL)** (Sec. 3.3), built around **TAURA** (Turn-aware Adaptive Uncertainty Rescaled Advantage). We identify that standard Group Relative Policy Optimization (GRPO) (Guo et al., 2025), when applied to multi-turn agentic reasoning, suffers from *Advantage Homogenization*, where a uniform trajectory-level advantage conflates pivotal discovery turns with trivial actions. TAURA resolves this by using turn-level entropy to steer credit assignment

toward these critical moments. Our main contributions are summarized as follows:

- **First Native Omni-Modal Active Perception Framework:** We introduce **OmniAgent**, which, to our knowledge, is the first end-to-end native agentic framework that unifies perception, reasoning, and action within a single model for omni-modal video tasks. By formulating multimodal exploration as a POMDP with a persistent textual memory, OmniAgent effectively decouples reasoning complexity from video duration, enabling scalable reasoning over hour-long videos.

- **Two-Stage Agentic Optimization:** We propose a two-stage optimization comprising (i) **Agentic SFT**, which bootstraps native active perception through a best-of-N trajectory synthesis pipeline with dual-stage quality control, and (ii) **TAURA**, an entropy-steered RL objective that resolves advantage homogenization in GRPO by using turn-level entropy to amplify credit for pivotal discovery turns over trivial actions.

- **Comprehensive SoTA Performance: OmniAgent** establishes new state-of-the-art results among open-source models across **ten benchmarks**, improving over the direct Qwen2.5-Omni baseline on all of them. It advances long-video comprehension (LVBench 50.5%, MLVU 71.1%), excels in omni-modal understanding

(DailyOmni +4.7%, OmniVideo +7.8%), and achieves an absolute +33.4% gain in temporal grounding on LongVALE. Notably, our 7B agent outperforms the $10\times$ larger Qwen2.5-VL-72B on LVBench (50.5% vs. 47.3%) with 73% fewer frames, and demonstrates positive *test-time scaling* (+6.2% on VideoMME-Long).

## 2. Related Work

### 2.1. Passive Omni Modality Understanding

The advancement of Omni Large Language Models (OmniLLMs) has been marked by significant proprietary contributions like GPT-4o (OpenAI, 2024) and Project Astra (Google DeepMind, 2024), alongside a surge in open-source models (Fu et al., 2024; Li et al., 2024; Cheng et al., 2024; Xu et al., 2025a; Xing et al., 2025; Tang et al., 2025; Yang et al., 2025b; Xu et al., 2025b; Meituan LongCat Team, 2025). While these initiatives enhance audio-visual understanding, existing passive methods struggle with long contexts due to the prohibitive complexity of continuous signals. Unlike static frame sampling, the natural temporal continuity of audio-visual streams hinders global simultaneous attention. To address this, OmniAgent endows MLLMs with native active perception, leveraging video compositionality to effectively decouple reasoning complexity from the video duration.

### 2.2. Agentic Video Reasoning

Pioneered by VisProg (Gupta & Kembhavi, 2023), agentic video understanding has evolved along two primary trajectories. The first leverages LLMs to orchestrate expert modules, relying on pre-extracted contexts like captions (Wang et al., 2024) and summaries (Wang et al., 2023; Ma et al., 2025), structured planning (Yang et al., 2025c; 2024; Wang et al., 2025d), or diverse tools including tracking (Fan et al., 2024), ASR (Tao et al., 2025), search (Zhang et al., 2025b; Long et al., 2026), and ensembles (Chen et al., 2025a). The second adapts "Think with Image" paradigms to video, utilizing transformations like temporal clipping (Zhang et al., 2026; Yang et al., 2026), spatial zooming (Shen et al., 2025a), or combinatorial cropping (Rasheed et al., 2026) for exhaustive analysis. However, both approaches neglect video's inherent sequential nature, treating it effectively as a static information container or large image, which hampers scaling. To address this, we formulate video understanding as a Partially Observable Markov Decision Process (POMDP), injecting native agentic capabilities into MLLMs to achieve robust test-time scaling without reliance on external modules.

## 3. OmniAgent

OmniAgent reconceptualizes omni-modal video reasoning by bridging the gap between perception and cognition, treating **active perception** not as a preprocessing step but as

---

**Algorithm 1** OmniAgent: Active Perception and Memory Consolidation

**Require:** Query $Q$, metadata $V_{\text{meta}}$, horizon $K$.
**Ensure:** Terminal answer $y$.

1: **Initialize:** Persistent memory $\mathcal{M}_0 \leftarrow \{Q, V_{\text{meta}}\}$, transient multimodal percept $\mathcal{E}_0 \leftarrow \emptyset$.

2: **for** $k = 1$ to $K$ **do**
3:     **PHASE 1: ACTIVE PERCEPTION**
4:       $(O_k, T_k, A_k) \sim \pi_\theta(\cdot \mid \mathcal{M}_{k-1}, \mathcal{E}_{k-1})$
5:       ▷ **Query-Conditional:** Grounds exploration in intent $Q$ and current context.

6:     **PHASE 2: MEMORY CONSOLIDATION**
7:       $\mathcal{M}_k \leftarrow \mathcal{M}_{k-1} \cup \{(O_k, T_k, A_k)\}$
8:       ▷ **Decoupling:** Maintains constant-order media overhead.

9:     **if** $A_k = a_{\text{answer}}(y)$ **then**
10:       **return** $y$
11:     **end if**

12:     **PHASE 3: PERCEPTUAL TRANSITION**
13:       $\mathcal{E}_k \leftarrow \Omega(A_k)$
14:       ▷ Environment **executes** $A_k$, **purging** previous $\mathcal{E}_{k-1}$.
15: **end for**

---

a query-driven reasoning process. Unlike passive models that process inputs uniformly, our framework decouples reasoning from multimodal data by establishing a strict separation between transient multimodal percept and persistent textual memory. This architecture enables the agent to selectively distill critical audio-visual cues into a compact textual memory while discarding high-dimensional raw media (see Figure 1). Consequently, OmniAgent maintains a long-horizon reasoning trace where the contextual cost is largely independent of the video duration. OmniAgent is optimized through a two-stage regime: (i) **Agentic SFT** (Sec. 3.2) to bootstrap fundamental action execution capabilities, and (ii) **Agentic RL** (Sec. 3.3) to refine reasoning-driven perception, with TAURA mitigating the credit assignment challenge in multi-turn reasoning.

### 3.1. Overall Agentic Pipeline

We formulate the interaction between OmniAgent and the video environment as a Partially Observable Markov Decision Process (POMDP). Here, the transient percept $\mathcal{E}_k$ represents the raw media returned by the environment $\Omega$, while the persistent memory $\mathcal{M}_k$ serves as the agent's consolidated internal state. Unlike passive models that process video frames uniformly, OmniAgent employs a policy $\pi_\theta$ to selectively distill the transient $\mathcal{E}_{k-1}$ into a compact textual observation $O_k$.

The process executes through an iterative *Observation-Thought-Action (OTA)* cycle (Algorithm 1). The state

at turn $k$ is formalized as the memory $\mathcal{M}_k = (O_0, \text{OTA}_1, \ldots, \text{OTA}_k)$, where initial state $\mathcal{M}_0 = \{Q, V_{\text{meta}}\}$ contains the query and video metadata (e.g., duration, FPS, audio availability). At each turn $k \in \{1, \ldots, K\}$, where $K$ is the maximum turn limit, the agent generates the OTA triplet autoregressively. The policy conditions on both the persistent memory and the transient percept:

$$(O_k, T_k, A_k) \sim \pi_\theta(\cdot \mid \mathcal{M}_{k-1}, \mathcal{E}_{k-1}) \quad (1)$$

At $k = 1$ ($\mathcal{E}_0 = \emptyset$), the policy conditions solely on $\mathcal{M}_0$ to bootstrap the initial exploration (see Appendix A for a complete notation summary).

**Observation ($O_k$):** A structured textual summary that distills the high-dimensional percept $\mathcal{E}_{k-1}$ into persistent memory. Unlike raw pixels, $O_k$ serves as an information-dense encoding for the reasoning process, explicitly retaining critical visual and auditory details required for future reasoning before the raw media is purged.

**Thought ($T_k$):** The internal reasoning process that bridges perception and action. $T_k$ analyzes the preceding memory $\mathcal{M}_{k-1}$ and the current observation $O_k$ to reason over the accumulated evidence. It identifies information gaps between the current percept and the query requirements, deriving the rationale for the subsequent action $A_k$.

**Action ($A_k$):** The symbolic operator sampled from $\mathcal{A} = \{a_{\text{frames}}, a_{\text{audio}}, a_{\text{clip}}, a_{\text{answer}}\}$. Specifically: $a_{\text{frames}}(s, e, n)$ retrieves $n$ frames uniformly from the time interval $[s, e]$, offering flexible temporal resolution; $a_{\text{audio}}(s, e)$ extracts the audio segment; $a_{\text{clip}}(s, e)$ captures a continuous video segment with *synchronized audio* to preserve temporal continuity and cross-modal alignment; and $a_{\text{answer}}(y)$ emits the final answer $y$, terminating the trajectory.

**Memory Consolidation.** The environment $\Omega$ facilitates turn-level transitions by resolving $A_k$ into new percept $\mathcal{E}_k$. This triggers a strict *context purging* mechanism: the previous raw multimodal percept $\mathcal{E}_{k-1}$ is discarded from the active context, leaving only the distilled text $O_k$ in $\mathcal{M}_k$. This ensures the model's media overhead remains constant regardless of the video duration or the number of interaction turns (see Appendix B for context management details).

Note that $\Omega$ performs only raw media extraction (frame retrieval, audio extraction, clip capture); all semantic perception and reasoning are carried out natively by $\pi_\theta$ without relying on external modules.

## 3.2. Agentic Supervised Fine-Tuning

Directly optimizing OmniAgent via reinforcement learning risks policy collapse, as base models (Xu et al., 2025a) lack prior training for long-horizon agentic reasoning. To bootstrap these capabilities, we curate an Agentic SFT corpus of 58K trajectories across three task categories (MCQ, numerical reasoning, and temporal grounding), derived from the training splits of five datasets: LongVideo-Reason (Chen et al., 2025c), Video-Holmes (Cheng et al., 2025), VSI-Train-10k (Brown et al., 2025), LongVALE (Geng et al., 2025), and MultiHop-EgoQA (Chen et al., 2025b). This corpus is strictly aligned with the iterative $(O_k, T_k, A_k)$ cycle formalized in Algorithm 1.

**Synthesis via Exploration.** Instead of relying on static QA annotations, we prompt a teacher model (e.g., Gemini-3.0-Pro) via in-context learning with the instruction template in Appendix B.5 to perform success-driven exploration in the environment $\Omega$. For each query, we execute a *best-of-N generation* over the action space $\mathcal{A}$ to produce a diverse pool of candidate trajectories. This generation process explicitly allows for *self-correction*, where the model initially executes invalid actions (*e.g.*, out-of-bounds timestamps) but successfully recovers based on the symbolic environment feedback. Including these error-correction traces prevents the "teacher-forcing" bias, training OmniAgent to interpret diagnostic signals as actionable cues rather than fatal failures (see Appendix B for diagnostic error protocols).

**Dual-Stage Quality Control.** To distill high-quality training data from the raw candidate pool, we implement a two-step filtration pipeline. (1) **Outcome Verification:** We first filter for correctness based on the task-specific success criteria defined in Eq. 2. Specifically, we require *exact matches* for discrete tasks (MCQ, Numerical) and enforce threshold-based criteria for continuous tasks: Intersection over Union (IoU) $\geq 0.5$ for temporal grounding and Mean Relative Accuracy (MRA) $\geq 0.5$ for size estimation. (2) **Rationality Audit:** Since the textual memory decouples reasoning from raw media, we employ GPT-4o (OpenAI, 2024) to audit the *internal coherence* of the reasoning trace. GPT-4o evaluates whether the current Thought $T_k$ is logically entailed by the accumulated memory $\mathcal{M}_{k-1}$ and the immediate observation $O_k$ on a 5-point Likert scale. This step filters out "lucky guesses", trajectories that reach the correct answer through hallucinated or heuristic reasoning steps that are not supported by the recorded memory. By enforcing a minimum coherence score of 3/5, we ensure that all SFT actions are rationally grounded in the agent's explicit context.

## 3.3. Agentic Reinforcement Learning

To incentivize the policy to handle more complex interactions within the proposed environment and facilitate self-evolution, we further optimize OmniAgent using reinforcement learning. By utilizing verifiable rewards, we advance the model beyond the initial priors established during SFT.

**Verifiable Reward Design.** OmniAgent is optimized to maximize a task-specific verifiable reward $R$, quantifying

the alignment between the prediction $\hat{y}$ and ground-truth $y$:

$$R(\hat{y}, y) = \begin{cases} \mathbb{1}[\hat{y} = y] & \text{Discrete (MCQ / Numerical)} \\ \text{IoU}(\hat{y}, y) & \text{Temporal Grounding} \\ \text{MRA}(\hat{y}, y) & \text{Continuous (Size Estimation)} \end{cases} \quad (2)$$

where $\text{IoU}(\hat{y}, y) = \frac{|\mathcal{I}_{\hat{y}} \cap \mathcal{I}_y|}{|\mathcal{I}_{\hat{y}} \cup \mathcal{I}_y|}$ represents temporal overlap, and MRA assesses quantitative fidelity across relative precision thresholds $\mathcal{T} = \{0.5, \ldots, 0.95\}$.

**The Advantage Homogenization Problem.** Applying Group Relative Policy Optimization (GRPO) (Guo et al., 2025) to multi-turn agentic reasoning faces a structural limitation: *Advantage Homogenization*. By broadcasting a single scalar advantage to every turn, vanilla GRPO inherently overlooks the heterogeneous contributions of individual turns, conflating pivotal *forks* with trivial fillers. This flaw is substantiated by our empirical analysis in Appendix C, which reveals that 79.2% of critical branching turns exhibit significantly higher mean token entropy than the trajectory mean. Consequently, vanilla GRPO's uniform advantage broadcasting masks the causal significance of high-uncertainty discovery moments, necessitating the entropy-steered credit assignment in TAURA.

**TAURA: Entropy-Steered Credit Assignment.** To address the advantage homogenization problem, we propose **TAURA** (Turn-aware Adaptive Uncertainty Rescaled Advantage), which refines trajectory-level advantages into turn-level attributions. Our method is inspired by findings that high-entropy tokens often represent critical "forks" in reasoning (Wang et al., 2025b). However, while prior work suggests masking low-entropy tokens in a Chain-of-Thought, such a binary masking strategy is ill-suited for agentic trajectories. Since the atomic reasoning unit is the structured turn $(O_k, T_k, A_k)$, masking individual tokens disrupts the output structure, while masking entire turns severs essential contextual dependencies and semantic continuity.

Formally, for a group of $G$ trajectories sampled from the same query, we first compute the baseline trajectory-level advantage $A_i$ by normalizing the rewards:

$$A_i = \frac{R_i - \frac{1}{G} \sum_{j=1}^{G} R_j}{\text{std}(R_1, \ldots, R_G)} \quad (3)$$

where $R_i$ is the reward for the $i$-th trajectory as defined in Eq. 2.

Building upon this baseline, TAURA implements a turn-level rescaling. By using mean token entropy as a continuous weighting factor rather than a binary mask, TAURA prioritizes turns with higher information density while preserving gradient flow for all tokens. For each turn $k$ within trajectory $i$, let $H_{i,k}$ denote its mean token entropy. We

define the rescaled advantage $\hat{A}_{i,k}$ by normalizing $H_{i,k}$ relative to the group mean:

$$\hat{A}_{i,k} = A_i \cdot \underbrace{\frac{H_{i,k}}{\frac{1}{N_{\mathcal{G}}} \sum_{j=1}^{G} \sum_{m=1}^{K_j} H_{j,m}}}_{w_{i,k}} \quad (4)$$

where $G$ is the group size, $K_j$ denotes the turn count of trajectory $j$, and $N_{\mathcal{G}} = \sum_{j=1}^{G} K_j$ represents the total turn count across the group. This normalization ensures that the expected weight $\mathbb{E}[w_{i,k}] = 1$ over the group, thereby preserving the original gradient scale while directing updates toward high-uncertainty discovery moments.

The policy is subsequently optimized using the TAURA-enhanced GRPO objective. To resolve the symbol ambiguity, we denote the $i$-th trajectory as $\tau_i$ and its constituent tokens as $\{o_{i,t}\}_{t=1}^{|\tau_i|}$. The surrogate objective is formalized as:

$$\mathcal{J}(\theta) = \mathbb{E}_{q \sim \mathcal{D}, \{\tau_i\}_{i=1}^{G} \sim \pi_{\theta_{\text{old}}}} \left[ \frac{1}{G} \sum_{i=1}^{G} \frac{1}{|\tau_i|} \sum_{t=1}^{|\tau_i|} \mathcal{L}_{i,t}(\theta) \right] \quad (5)$$

where the per-token loss $\mathcal{L}_{i,t}(\theta)$ incorporates the turn-level rescaled advantage:

$$\mathcal{L}_{i,t}(\theta) = \min\big(\rho_{i,t} \hat{A}_{i,\text{turn}(t)}, \\ \text{clip}(\rho_{i,t}, 1 - \epsilon, 1 + \epsilon)\hat{A}_{i,\text{turn}(t)}\big) \quad (6)$$

Here, $\rho_{i,t} = \frac{\pi_\theta(o_{i,t}|q, o_{i,<t}, \mathcal{E}_{\text{turn}(t)-1})}{\pi_{\theta_{\text{old}}}(o_{i,t}|q, o_{i,<t}, \mathcal{E}_{\text{turn}(t)-1})}$ is the importance sampling ratio. The mapping function $\text{turn}(t)$ assigns each token $t$ to its corresponding turn index $k \in \{1, \ldots, K_i\}$, ensuring that the credit for each token is steered by the information density of its constituent turn.

**Why Entropy Scaling Works.** TAURA scales the *signed* advantage $A_i$. For correct trajectories ($A_i > 0$), high entropy ($w_{i,k} > 1$) amplifies the advantage, upweighting turns where the model navigated genuine uncertainty. Conversely, for incorrect trajectories ($A_i < 0$), high entropy results in a larger *negative* penalty ($\hat{A}_{i,k} < A_i < 0$). This strictly penalizes confused guessing while reinforcing valid discovery actions.

## 4. Experimental Results

### 4.1. Experimental Settings

**Benchmarks and Metrics.** We evaluate across ten benchmarks in three categories. (1) *Video Understanding*: VideoMME (generic) (Fu et al., 2025), VSI-Bench (reasoning) (Yang et al., 2025a), MLVU (long) (Zhou et al., 2025a), Minerva (reasoning & long) (Nagrani et al., 2025), and LVBench (long) (Wang et al., 2025c). (2) *Audio-Visual*:

*Table 1.* **Main results on video understanding and reasoning.** We evaluate OmniAgent across a comprehensive suite of benchmarks featuring diverse temporal scales. **Bold** and underline indicate the **best** and second-best performance among open-source models, respectively. Methods marked with ∗ incorporate audio signals. VISTA result is based on LongVA. Δ denotes the performance gain relative to the Qwen2.5-Omni baseline.

| Methods | Size | VideoMME (w/o sub.) | | VSI-Bench | MLVU | Minerva | LVBench |
|---|---|---|---|---|---|---|---|
| | | Overall | Long | AVG | M-AVG | AVG | AVG |
| *Duration* | | 1–60 min | 30–60 min | 97 sec | 3–120 min | 2–90 min | 4101 sec |
| *Proprietary Models* | | | | | | | |
| GPT-4o (OpenAI, 2024) | – | 71.9 | 65.3 | 34.0 | 64.6 | 45.5 | 48.9 |
| Gemini-1.5-Pro (Gemini Team, 2024) | – | 75.0 | 67.4 | 45.4 | – | – | 33.1 |
| Gemini-2.5-Pro (Comanici et al., 2025) | – | – | – | 51.5 | – | 66.2 | 67.4 |
| *Open-Source Agentic Models* | | | | | | | |
| LongVT (Yang et al., 2026) | 7B | 55.9 | 44.4 | 34.4 | – | 28.5 | 41.3 |
| Zoom-Zero (Shen et al., 2025a) | 7B | 66.0 | 54.8 | – | 70.8 | – | 45.7 |
| VITAL (Zhang et al., 2026) | 7B | 64.1 | 54.0 | 41.8 | – | – | – |
| Video-CoM (Rasheed et al., 2026) | 7B | 59.4 | – | – | 60.9 | 31.7 | – |
| *Open-Source Thinking Models* | | | | | | | |
| Video-R1 (Feng et al., 2025) | 7B | 61.4 | – | 37.1 | 60.9 | 29.1 | 40.1 |
| Open-o3 Video (Meng et al., 2026) | 7B | 63.6 | 54.9 | – | – | – | – |
| VideoRFT (Wang et al., 2025a) | 7B | 59.8 | – | 36.8 | 59.7 | 29.2 | 34.7 |
| LongVILA-R1 (Chen et al., 2025c) | 7B | 65.1 | 55.2 | – | – | – | – |
| *Open-Source Non-Thinking Models* | | | | | | | |
| Kangaroo (Liu et al., 2024) | 8B | 56.0 | 46.7 | – | 61.0 | – | 39.4 |
| VideoLLaMA2∗ (Cheng et al., 2024) | 7B | 47.9 | – | – | 48.5 | – | – |
| LongVA (Zhang et al., 2025a) | 7B | 51.8 | 46.1 | 29.2 | 56.3 | – | 35.9 |
| LLaVA-OneVision (Li et al., 2025) | 7B | 58.2 | 46.7 | 32.4 | 64.7 | – | – |
| VISTA (Ren et al., 2025b) | 7B | 55.5 | 47.4 | – | 62.1 | – | 39.0 |
| VideoChat-T (Zeng et al., 2025) | 7B | 46.3 | 41.9 | – | – | – | – |
| LongVILA (Chen et al., 2025d) | 7B | 60.1 | 53.0 | 21.6 | – | – | – |
| Vamba (Ren et al., 2025a) | 10B | 57.8 | – | – | 65.9 | – | 42.1 |
| LongVU (Shen et al., 2025b) | 7B | 60.6 | 59.5 | – | 65.4 | – | – |
| Qwen2.5-VL (Bai et al., 2025) | 7B | 65.1 | – | 33.5 | – | 33.0 | 45.3 |
| Qwen2.5-Omni∗ (Xu et al., 2025a) | 7B | 64.8 | 54.8 | 35.5 | 65.2 | 33.4 | 43.0 |
| **OmniAgent (Ours)**∗ | **7B** | **67.8** | **59.6** | **48.4** | **71.1** | **41.4** | **50.5** |
| Δ *over Baseline* | | +3.0 | +4.8 | +12.9 | +5.9 | +8.0 | +7.5 |

DailyOmni (generic) (Zhou et al., 2025b), WorldSense (generic) (Hong et al., 2025), and OmniVideoBench (reasoning) (Li et al., 2026). (3) *Temporal Grounding*: Long-VALE (Geng et al., 2025) and VUE-TR (Vidi Team, 2025). Together, these benchmarks cover video durations ranging from tens of seconds to over two hours. The evaluation metrics (MCQ accuracy, Temporal IoU, MRA) are aligned with the verifiable reward functions defined in Sec. 3.3.

**Implementation Details.** We adopt Qwen2.5-Omni-7B (Xu et al., 2025a) as the base model. To manage context length, the maximum number of visual tokens per image and per video frame is set to 1024 and 768, respectively, with a maximum context window of 64K tokens. Both training stages employ a dynamic maximum turn limit $K$ (Algorithm 1) that scales with video duration ($K \in [5, 32]$ for Agentic SFT, $K \in [5, 10]$ for Agentic RL).

The same agent instruction template (Appendix B.5) is used across Agentic SFT, Agentic RL, and inference. The Agentic SFT is conducted on 58K trajectories for 2 epochs, with a learning rate of $1 \times 10^{-5}$, a batch size of 64, and the AdamW optimizer on 16 NVIDIA A100 GPUs.

For Agentic RL, we specifically select queries from Sec. 3.2 where best-of-N sampling failed to yield successful trajectories, thereby focusing RL on exploring and resolving these challenging cases. All video durations for RL are restricted to under 300 seconds. Following DAPO (Yu et al., 2025a), we employ a token-level policy loss with the clip-higher mechanism. The model is optimized for 150 steps with a group size of 8, a constant learning rate of $1 \times 10^{-6}$, and upper/lower clip ratios of 0.30 and 0.20, respectively. The global batch size is 256 on 64 NVIDIA A100 GPUs, and neither KL nor entropy regularization is applied.

*Table 2.* **Main results on audio-visual understanding and reasoning.** OmniAgent is evaluated on video benchmarks requiring joint audio-visual reasoning. **Bold** and underline indicate the **best** and second-best performance among open-source models, respectively. Δ denotes the performance gain relative to the Qwen2.5-Omni baseline.

| Models | Size | DailyOmni | WorldSense | OmniVideo |
|---|---|---|---|---|
| | | AVG | AVG | AVG |
| *Duration* | | 43 sec | 141 sec | 384 sec |
| *Proprietary Models* | | | | |
| GPT-4o (OpenAI, 2024) | – | 56.5 | 42.6 | – |
| Gemini-1.5-Pro (Gemini Team, 2024) | – | – | 48.0 | – |
| Gemini-2.0-Flash (Gemini Team, 2023) | – | 56.1 | – | 41.5 |
| *Open-Source Models* | | | | |
| Unified-IO-2 (Lu et al., 2024) | 8B | 28.2 | 25.9 | – |
| VideoLLaMA2 (Cheng et al., 2024) | 7B | 35.2 | 25.4 | 29.2 |
| MiniCPM-o (Yu et al., 2025b) | 8B | 53.1 | – | 29.7 |
| Ola (Liu et al., 2025) | 7B | 50.7 | – | – |
| Qwen2.5-Omni (Xu et al., 2025a) | 7B | 60.1 | 45.4 | 29.3 |
| **OmniAgent (Ours)** | **7B** | **64.8** | **47.2** | **37.1** |
| Δ *over Baseline* | | +4.7 | +1.8 | +7.8 |

*Table 3.* **Main results on audio-visual temporal grounding.** We report temporal IoU score on LongVALE and VUE-TR. OmniAgent achieves state-of-the-art results among both proprietary and open-source models. **Bold** and underline indicate the **best** and second-best performance among open-source models, respectively. Δ denotes the performance gain relative to the Qwen2.5-Omni baseline.

| Models | Size | LongVALE | VUE–TR | |
|---|---|---|---|---|
| | | IoU | Vision+Audio | Vision |
| *Duration* | | 233 sec | 1066 sec | 1114 sec |
| *Proprietary Models* | | | | |
| GPT-4o (OpenAI, 2024) | – | – | 11.1 | 22.0 |
| Gemini-2.0-Flash (Gemini Team, 2023) | – | – | 18.4 | 25.4 |
| Gemini-2.5-Pro (Comanici et al., 2025) | – | – | 12.1 | 21.8 |
| *Open-Source Agentic Models* | | | | |
| VITAL (Zhang et al., 2026) | 7B | – | – | 35.3 |
| *Open-Source Models* | | | | |
| LongVALE-LLM (Geng et al., 2025) | 7B | 11.0 | – | – |
| Vidi (Vidi Team, 2025) | 7B | – | 33.4 | 43.9 |
| Qwen2.5-Omni (Xu et al., 2025a) | 7B | 5.7 | 3.5 | 8.0 |
| **OmniAgent (Ours)** | **7B** | **39.1** | **36.5** | **46.1** |
| Δ *over Baseline* | | +33.4 | +33.0 | +38.1 |

## 4.2. Main Results

We evaluate OmniAgent across three task categories. Results are summarized in Tables 1, 2, and 3.

**Video Understanding and Reasoning.** OmniAgent-7B achieves state-of-the-art performance among open-source models on long-video benchmarks (Table 1). On LVBench, it scores 50.5%, substantially surpassing the Qwen2.5-Omni-7B baseline (43.0%) listed in Table 1. Notably, it even outperforms the 10× larger Qwen2.5-VL-72B (47.3%; see Figure 3) (Bai et al., 2025) and the agentic baseline Zoom-Zero-7B (45.7%) (Shen et al., 2025a), while using far fewer frames. Furthermore, OmniAgent outperforms recent "Thinking Models" like Video-R1 (60.9% on MLVU) (Feng et al., 2025). While thinking models rely on extended Chain-of-Thought (CoT) reasoning over static inputs, OmniAgent (71.1% on MLVU) actively queries the environment to retrieve missing evidence. This suggests that for long-form video, the primary bottleneck is often perceptual incompleteness rather than reasoning depth. Compared to the passive baseline LongVU (Shen et al., 2025b) which uses dense sampling (1 FPS), OmniAgent achieves higher accuracy on VideoMME (+7.2%) and MLVU (+5.7%), validating that query-conditional active perception is more sample-efficient than uniform processing.

**Audio-Visual Understanding and Reasoning.** On benchmarks requiring joint perception (Table 2), OmniAgent outperforms the native omni-modal baseline Qwen2.5-Omni (Xu et al., 2025a) on DailyOmni (+4.7%) and OmniVideoBench (+7.8%). Passive models process modalities in a single pass, often trading off resolution for context. In contrast, OmniAgent utilizes auditory cues as temporal anchors to guide subsequent visual sampling, converting audio events into targeted visual search queries.

**Temporal Grounding.** OmniAgent shows substantial improvements in temporal localization (Table 3), achieving absolute gains over Qwen2.5-Omni on LongVALE (+33.4%) and VUE-TR (+33.0%). Notably, OmniAgent-7B surpasses proprietary models including GPT-4o and Gemini-2.5-Pro on VUE-TR. This performance stems from our iterative strategy: rather than regressing coordinates from a compressed global view, the agent employs on-demand sampling to progressively narrow the search space from coarse to fine granularity, ensuring higher precision.

**Qualitative Analysis.** To illustrate the interpretability of our framework, we visualize reasoning trajectories in Figure 1 and Appendix D. Figure 1 exemplifies query-conditional perception, where the agent leverages the temporal constraint ("22:03") from the user query to steer its exploration, ignoring irrelevant segments. Extended case studies (Figures 6, 7, and 8) further demonstrate how the agent bridges information gaps through active audio-visual interactions across both reasoning and grounding tasks.

## 4.3. Ablation Study and Analysis

**Impact of Training Paradigms.** Table 4 presents the impact of shifting from passive to active perception. (1) *Standard SFT*: Fine-tuning on static QA pairs induces a performance regression on extremely long contexts (LVBench 43.0% → 41.6%). Lacking a selection mechanism, the passive paradigm suffers from information overload, where redundant frames reduce the signal-to-noise ratio as duration increases. (2) *Agentic SFT*: In contrast, Agentic SFT substantially improves long-video comprehension on LVBench (48.7%) and omni-modal understanding on Daily-Omni (60.1% → 63.3%), demonstrating the effectiveness of the $(O_k, T_k, A_k)$ formulation.

*Table 4.* **Component Ablation Study.** We compare training paradigms and RL algorithms. Agentic SFT establishes the baseline for active perception, while TAURA mitigates the advantage homogenization problem of Vanilla GRPO, improving both understanding and reasoning tasks.

| Method | Video Understanding | | | Omni-Modal Reasoning | |
|---|---|---|---|---|---|
| | VideoMME$_{Long}$ | LVBench | MLVU | OmniVideo | DailyOmni |
| Qwen2.5-Omni | 54.8 | 43.0 | 65.2 | 29.3 | 60.1 |
| + Standard SFT | 56.0 | 41.6 ↓ | 67.1 | 33.8 | 61.7 |
| + Agentic SFT | 57.3 | 48.7 | 69.9 | 35.2 | 63.3 |
| *RL Stage (Initialized from Agentic SFT)* | | | | | |
| + Vanilla GRPO | 59.4 | 49.8 | 69.9 | 35.3 | 62.2 ↓ |
| **+ TAURA** | **59.6** | **50.5** | **71.1** | **37.1** | **64.8** |

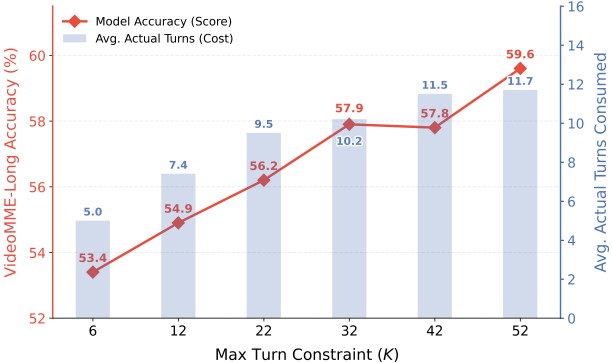

*Figure 2.* **Test-time scaling on VideoMME-Long.** Accuracy improves with the maximum turn limit ($K$), demonstrating positive scaling. The average number of turns executed saturates at 11.7 even when $K = 52$, indicating that the model adaptively adjusts its reasoning depth based on information need rather than maximizing the turn count.

**Efficacy of TAURA vs. Vanilla GRPO.** (1) *Advantage Homogenization*: Vanilla GRPO stagnates on reasoning (MLVU 69.9%) and degrades perception (DailyOmni 63.3% → 62.2%). By broadcasting a uniform trajectory-level advantage, it fails to incentivize specific perceptual discoveries, as critical "looking" actions receive the same credit as trivial actions. (2) *Entropy-Steered Assignment*: TAURA rectifies this by using entropy as a proxy for decision criticality. By up-weighting high-entropy turns—thereby identifying decisive "forks" in the search space—TAURA aligns credit assignment with information gain. This fine-grained supervision drives consistent improvements across both perception (DailyOmni 64.8%) and reasoning (MLVU 71.1%).

**Test-time Scaling Analysis.** We analyze the scaling properties on VideoMME-Long by extending the maximum turn limit $K$ from 6 to 52. Figure 2 reveals a monotonic performance improvement, where accuracy climbs from 53.4% to 59.6% (+6.2%). This confirms that deeper interaction enables the agent to uncover critical evidence for complex queries. Even as the upper bound expands by nearly 9×,

*Table 5.* **Duration Analysis on LVBench.** As video duration increases, the agent maintains stable accuracy despite a substantial drop in sampling density. This demonstrates that computational cost is driven by task complexity, not video duration.

| Duration (min) | 20–40 | 40–60 | 60–80 | 80–100 | 100–120 | 120–140 |
|---|---|---|---|---|---|---|
| Count | 309 | 420 | 278 | 302 | 173 | 67 |
| **Avg. Turns** | 8.5 | 9.9 | 11.3 | 11.2 | 10.8 | 12.5 |
| **Turns / Hour** | 16.9 | 11.9 | 9.7 | 7.5 | 5.9 | 5.7 |
| **Accuracy (%)** | 53.7 | 52.1 | 45.3 | 48.0 | 53.2 | 50.8 |

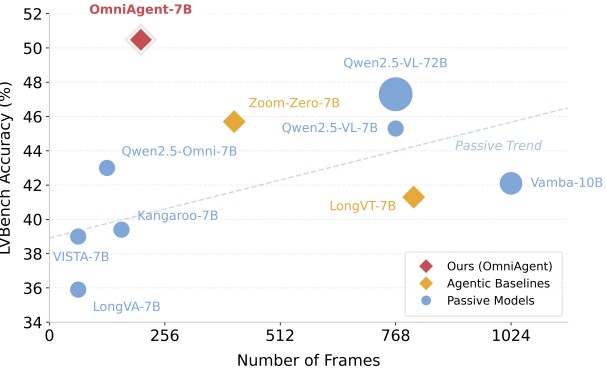

*Figure 3.* **Accuracy vs. Visual Frame Count on LVBench.** OmniAgent-7B (red diamond, 50.5%) outperforms the 10× larger Qwen2.5-VL-72B (47.3%) while using ~73% fewer frames (203 vs. 768). Marker shape distinguishes agentic (diamond) from passive (circle) models, and marker size represents parameter scale.

the *actual average turns* saturate at merely ~11.7. This indicates that the agent does not blindly exhaust the available budget, but emits its final answer via $a_{answer}$ once sufficient evidence is gathered, terminating the trajectory. This saturation suggests that the reasoning depth is driven by query complexity rather than the maximum turn limit.

**Visual Sampling Efficiency.** Figure 3 illustrates the trade-off between accuracy and sampling cost, revealing a distinct Pareto efficiency for OmniAgent. (1) *Parameter Efficiency (7B against 72B)*: OmniAgent-7B achieves peak accuracy (50.5%) while sampling only 203 frames on average. In sharp contrast, the 10× larger Qwen2.5-VL-72B (47.3%) relies on a dense input of 768 frames. This demonstrates that active perception acts as a computation multiplier: by selectively filtering noise, a 7B agent can achieve higher information density than a 72B passive model processing raw streams. (2) *Contrast with Agentic Baselines*: Unlike baselines like Zoom-Zero-7B (45.7%) that incur a fixed "entry cost" (256 frames) for initial scanning, OmniAgent operates on a strictly on-demand basis. This eliminates redundant processing, showing that iterative, query-conditional search is more efficient than global scanning. We further report inference runtime in Appendix C.4.

**Impact of Video Duration.** Table 5 validates that OmniAgent effectively decouples reasoning complexity from video length. As video duration grows over $4\times$ ($\sim$30 to $\sim$130 min), the absolute reasoning turns grow only marginally (8.5 $\rightarrow$ 12.5 turns), causing the sampling density to drop sharply from 16.9 to 5.7 turns per hour. Crucially, accuracy remains stable (50.8%) even at the lowest density. This confirms that the agent ignores redundant content ("the haystack") to focus solely on critical evidence ("the needle"), allocating compute based on information density rather than temporal duration.

## 5. Conclusion

We present OmniAgent, which establishes active perception as an intrinsic reasoning process for omni-modal video understanding. Our central insight is that treating perception as iterative, query-driven information distillation, rather than exhaustive preprocessing, allows reasoning complexity to be driven by task difficulty rather than video duration. By bootstrapping active perception via Agentic SFT and refining it via Agentic RL with TAURA, we further show that native agentic capabilities can emerge within a single multimodal model without relying on external perception modules. Empirically, a 7B agent with active perception outperforms a $10\times$ larger passive model while consuming far fewer frames, and exhibits positive test-time scaling where additional reasoning steps yield consistent gains. Beyond accuracy, the explicit Observation-Thought-Action (OTA) cycle provides interpretable reasoning traces, offering a transparent alternative to black-box video reasoning. While the sequential interaction loop introduces latency overhead, future work will investigate parallelized exploration to address this constraint.

## Acknowledgement

The work described in this paper was supported in part by the Research Grants Council of the Hong Kong Special Administrative Region, China, under Project CUHK 14202125 and Project CUHK 14200824.

## Impact Statement

This paper presents work whose goal is to advance the field of Machine Learning. Our query-driven active perception mechanism promotes efficient use of audio-visual information, processing only task-relevant content rather than entire video streams. As with any video understanding system, potential risks include misuse for unauthorized surveillance or biased content analysis. The interpretable reasoning traces produced by our framework can help identify and mitigate such risks in downstream deployment.

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

## A. Detailed Mathematical Notation

To ensure clarity across the multi-level hierarchy of the OmniAgent framework, we provide a structured summary of the mathematical notations used throughout the paper. The framework involves three primary scales: (i) the trajectory level ($i$), (ii) the interaction turn level ($k$), and (iii) the autoregressive token level ($t$).

*Table 6.* Summary of mathematical notations used in the OmniAgent framework.

| Symbol | Description |
|---|---|
| *Interaction Hierarchy* | |
| $i, j$ | Indices for trajectories. $i$ usually denotes the specific trajectory being optimized, while $j$ is used as a summation index over the group $G$. |
| $k, m$ | Indices for interaction turns. $k$ usually denotes the current turn, while $m$ is used as a summation index over the total horizon $K_j$. |
| $t$ | Index for discrete tokens within an autoregressive sequence. |
| $K$ | Maximum number of interaction turns permitted (horizon). |
| $G$ | Group size for relative advantage computation (group size in GRPO). |
| $N_{\mathcal{G}}$ | Total number of turns across all trajectories in a group ($\sum_{j=1}^{G} K_j$). |
| *Agent-Environment States* | |
| $\mathcal{E}_k$ | **Transient Percept**: Raw multimodal data acquired from the environment $\Omega$ at turn $k$. |
| $\mathcal{M}_k$ | **Persistent Memory**: The cumulative textual reasoning trace up to turn $k$. |
| $O_0$ | Initial persistent memory state containing the query $Q$ and video metadata. |
| $O_k, T_k, A_k$ | The Observation, Thought, and Action triplet generated at turn $k$. |
| $\Omega$ | The interactive environment that resolves action $A_k$ into new percept $\mathcal{E}_k$. |
| $\pi_\theta$ | The agent policy parameterized by $\theta$. |
| *Reinforcement Learning (TAURA)* | |
| $R_i$ | Final reward assigned to trajectory $i$ based on outcome verifiability. |
| $A_i$ | **Trajectory-level Advantage**: Normalized relative reward for trajectory $i$ within its group. |
| $H_{i,k}$ | Mean token entropy for the sequence generated during turn $k$ of trajectory $i$. |
| $w_{i,k}$ | **Entropy Weight**: The rescaled importance factor for turn $k$ based on information density. |
| $\hat{A}_{i,k}$ | **Turn-aware Advantage**: The TAURA-rescaled advantage applied to turn $k$ of trajectory $i$. |
| $\rho_{i,t}$ | Importance sampling ratio for token $t$ in trajectory $i$. |
| $\mathrm{turn}(t)$ | Mapping function that assigns token $t$ to its corresponding turn index $k$. |

**Index Mapping and Dependency.** The policy $\pi_\theta$ conditions its generation at turn $k$ on the distilled memory $\mathcal{M}_{k-1}$ and the *previous* transient percept $\mathcal{E}_{k-1}$. The internal logic of the framework ensures that raw media percept $\mathcal{E}_{k-1}$ is purged from the active context window once it is distilled into the textual observation $O_k$, maintaining context efficiency while preserving the causal chain via $\mathcal{M}_k$. In the optimization phase, TAURA ensures that the global success signal $A_i$ is steered toward the specific turns $k$ where discovery (high entropy $H_{i,k}$) occurred.

## B. Agentic Audio-Visual Interaction Environment

This section provides the comprehensive implementation details of the environment $\Omega$, focusing on the distributed infrastructure and robust media processing protocols required for reproducibility.

### B.1. Distributed Architecture via Ray and Verl

To ensure computational efficiency and prevent memory bottlenecks (OOM) during large-scale RL training, the environment is implemented using a distributed actor-based architecture powered by **Ray** and integrated with the **Verl** framework.

- **Global Resource Singleton:** We utilize a detached Ray actor (specifically named `GlobalProcessor` in our codebase) to manage heavy initialization tasks. This actor ensures that tokenizer weights and multimodal processor configurations are loaded into memory exactly once per physical node, significantly reducing the startup latency of concurrent worker processes.

- **Parallel Worker Pool:** The `VideoQAMultiEnv` orchestrates a pool of remote workers, each maintaining an independent instance of the `SingleVideoQAEnv`. This design allows for asynchronous perception and trajectory generation parallelized across CPU cores.

## B.2. Robust Perception Operators (FFmpeg Implementation)

The environment resolves sensing actions $A_k$ into transient percept $\mathcal{E}_k$ using specialized `ffmpeg` operators. We employ a **two-stage seeking strategy** (coarse seeking via keyframes followed by accurate decoding) to ensure sub-second precision.

- **Visual Sampling ($a_{\mathbf{frames}}$):** Frames are extracted using `ffmpeg` with `-q:v 2` to maintain high visual fidelity. For queries near video boundaries, the environment employs an `-sseof` fallback mechanism to automatically adjust retrieval windows, preventing failures at terminal timestamps.

- **Auditory Retrieval ($a_{\mathbf{audio}}$):** Segments are decoded into a standardized **PCM S16LE** mono-channel WAV format (16kHz). We enforce strict duration validation via `ffprobe` to ensure the retrieved audio waveform strictly aligns with the agent's requested interval $[s, e]$.

- **Adaptive Clip Capture ($a_{\mathbf{clip}}$):** Sub-clips are encoded using the `libx264` codec with a Constant Rate Factor (`CRF`) of 20. To ensure interaction continuity under high system load, the environment implements a **progressive fallback strategy**: it prioritizes the `superfast` preset but automatically downgrades to `ultrafast` if encoding latency exceeds the timeout threshold.

## B.3. Exploration Incentives via Randomization

To improve policy robustness during Agentic SFT, we enforce randomized physical constraints sampled from discrete uniform distributions:

- **Discrete Parameter Grid:** Frame counts are sampled from $n \in [30, 60]$ (step size 2); clip durations from $d_{\mathrm{clip}} \in [30, 60]$ seconds (step size 2); and audio durations from $d_{\mathrm{audio}} \in [150, 300]$ seconds (step size 10).

- **Diagnostic Error Protocols:** $\Omega$ provides structured symbolic feedback (e.g., `Err.TS_OOB`, `Err.INVALID_JSON`). These signals allow the agent to acknowledge mistakes in the persistent memory $\mathcal{M}_k$ and execute self-correction in subsequent turns.

## B.4. Memory Consolidation and History Purging

As formalized in Algorithm 1, the environment manages the persistent memory $\mathcal{M}_k$ by purging media content from the interaction history. To decouple memory costs from video duration, the environment performs a non-destructive rewrite of the conversation trace:

1. **Media Removal:** Once a sensing turn $k$ is completed and its textual distillation $O_k$ is recorded, the environment iterates through the previous chat history. For every user message containing multimodal content, the environment **deletes** the dictionary entries of type `image`, `video`, or `audio`.

2. **Metadata-Preserving Rewrite:** To maintain the semantic integrity of the reasoning trace, the deleted media objects are replaced with a text-based summary appended with the marker `[MEDIA OMITTED - Refer to Observation` $O_k$ `]`:
   - For **Visual Sampling** ($a_{\mathrm{frames}}$), the environment recompiles the exact timestamps of all purged frames from the prior message content (e.g., *"Frames 10.0s-20.0s. Timestamps: [10.00s, 12.50s, 15.00s] [MEDIA OMITTED...]"*).
   - For **Audio and Video Clips** ($a_{\mathrm{audio}}$, $a_{\mathrm{clip}}$), the original headers containing the requested temporal intervals are preserved (e.g., *"Audio 230.00s-280.00s [MEDIA OMITTED...]"* or *"Clip 45.00s-60.00s [MEDIA OMITTED...]"*).

## B.5. Agent Instruction Template

Figure 4 presents the complete instruction template that governs agent-environment interaction across all stages of the OmniAgent pipeline, including trajectory synthesis (Sec. 3.2), reinforcement learning (Sec. 3.3), and inference. Runtime parameters (`max_steps`, `max_frames_len`, `max_audio_len`, `max_clip_len`) are populated according to the configuration in Sec. 4.

```
You are the Deep-Omni-Research Agent, a specialized multi-modal analyst for temporal forensic investigation. Your goal is to solve
    complex queries by meticulously inspecting video and audio data through a step-by-step "Observe-Think-Action" loop.

============== GLOBAL OPERATING RULES ==============
- META-Validation: The first message provides "Video META" (duration, fps, has_audio). Validate every timestamp against these limits.
- Audio Constraint: If has_audio is false, the get_audio action is FORBIDDEN. Skip audio analysis and rely on visual cues only.
- Media Persistence: Once media is returned, it becomes a TEXT PLACEHOLDER in the next turn.
  * Frame Placeholder Example: "Frames 10.00s-12.00s (num=5). Timestamps: [10.00s, 10.50s, 11.00s, 11.50s, 12.00s] [MEDIA OMITTED –
    Refer to your Observation]"
  * AUDIO/CLIP Placeholder Example: "Audio 10.00s-20.00s [MEDIA OMITTED – Refer to your Observation]"
- The "Memory" Requirement: Your observation must be an exhaustive, high-fidelity log. Once media is omitted, you will "forget" any
    detail not recorded here.
- Strategic Efficiency: DO NOT request the exact same action and range twice. However, you are encouraged to re-inspect important
    ranges via different modalities (e.g., get_clip after get_audio) or higher density (Zooming in) to extract NEW forensic details.
- Strict Fidelity: You MUST use exact timestamps (including decimals) provided in environment labels (e.g., 481.84s). Never round or
    approximate numbers.
- Evidence Traceability: You MUST prefix findings with the Full Evidence ID (e.g., "[Frames 10.0s-12.0s (num=5)]") in both observation
    and think fields.
- Environment Feedback: Pay attention to [ERROR] and [NOTICE] (remaining steps). Adjust your strategy immediately.

========== STRATEGIC INSPECTION GUIDELINES ==========
1. Visual Search (get_frames): (Max {max_frames_len} frames).
   - Scanning: Use wide ranges (e.g., start=0, end=duration, num={max_frames_len}) to discover the overall timeline and identify key
     milestones or potential scene cuts.
   - Precision: Use narrow windows (1-2s) with high num for micro-details (logos, text, fast motions, or subtle object state changes).
2. Counting & Re-ID: Assign approximate spatial locations [y, x] (0-100 scale; [0,0] is top-left) to each unique instance (e.g., "
   Person_A at [20, 45]") in your observation. This spatial ID prevents re-counting the same object across different frames/steps.
3. Temporal Bisection: Find 'start' and 'end' boundary frames where a state changes, then iteratively narrow the interval to locate the
   exact transition second or frame.
4. Audio Analysis (get_audio): (Max {max_audio_len}s).
   - Verbatim Logging: Identify speakers and transcribe speech near-verbatim. CRITICAL: Do not paraphrase or infer words to fit your
     hypothesis.
   - Acoustic Context: Identify critical off-screen or background sounds (e.g., footsteps, sirens, clicks) that provide environmental
     clues for temporal reasoning.
5. Multi-Modal Action Analysis (get_clip): (Max {max_clip_len}s).
   - Action & Temporal Dynamics: Analyze the nature of movement (speed, direction, continuity) and precise sequencing to solve "Who
     moved first?" or "Was the motion deliberate?".
   - Process Logic: Use when the continuous process of a state change (e.g., an object falling) is more critical than discrete start/end
     points.
   - Audio-Visual Synergy (Conditional): If has_audio is true, perform high-fidelity forensic matching (Sync, Active Speaker ID,
     Causality with time-lag).

===================== ACTIONS =====================
Exactly ONE action per turn in valid JSON:
1. {"type": "get_frames", "start": float, "end": float, "num": int}
2. {"type": "get_audio", "start": float, "end": float}
3. {"type": "get_clip", "start": float, "end": float}
4. {"type": "answer", "content": "string"}
   - MCQ: Letter only (e.g., "A").
   - TR: JSON array of one or more pairs, e.g., "[[10.5, 20.0], [35.0, 40.0]]".
   - NUM/SIZE: A single number string, e.g., "10.3".
   - FF: Detailed descriptive text.

============= STRICT EXECUTION PROTOCOL =============
- Forensic Rigor: Answering incorrectly is a failure. Rule out every possible distractor before concluding.
- The Confidence Gatekeeper: You MUST include a numeric confidence field (0.0-1.0) as a top-level JSON key. This represents your
    assessment of whether the evidence is sufficient to conclude.
- The "0.9" Behavioral Rule: You should only initiate the "answer" action when your confidence is >= 0.9. If it is lower, continue
    gathering evidence unless [NOTICE] indicates "FINAL STEP".
- Evidence Contradiction: In your think field, actively look for evidence that disproves your current leading hypothesis.
- Deadline Management: In "FINAL STEP", bypass the 0.9 threshold and provide your best-informed answer immediately.

=================== OUTPUT SCHEMA ===================
The response must contain ONLY the JSON object itself. Any text outside the curly braces ({})--including thoughts, explanations, or
    markdown fences (```json)--is strictly forbidden and will result in system failure.

{"observation": "[Clip 00.00s-00.00s] (T: 00.00s)[Obj_A at y,x] visual_detail. [Audio 00.0s-00.0s] exact_audio_log. [Key Fact]:
    forensic_finding.", "think": "Evidence Review: [Clip 00.00s-00.00s] confirms_or_contradicts [Frames 00.00s-00.00s (num=0)]. Gap
    Analysis: missing_or_ambiguous_details. Deduction: logical_path_to_action_or_answer.", "confidence": 0.0, "action": {"type": "
    get_frames|get_audio|get_clip|answer", "start": 0.0, "end": 0.0, "num": 0, "content": ""}}

============= CRITICAL FORMATTING RULES =============
- Physical Boundary: Your entire response MUST start with '{' and end with '}' exactly.
- The "One-Line" Mandate: Your entire output MUST be ONE single line of text. NO newlines (\n) allowed anywhere.
- NO Markdown: Output raw text ONLY. DO NOT use code blocks or wrappers.
```

*Figure 4.* The complete agent instruction template used across all stages of the OmniAgent pipeline.

## C. Empirical Analysis: Entropy as a Proxy for Reasoning Criticality

To validate the motivation behind TAURA, we conducted a detailed analysis of the agent's reasoning traces to quantify the relationship between model uncertainty (entropy) and reasoning criticality.

### C.1. Methodology: Identifying Decision Forks

We hypothesize that in a multi-turn agentic trajectory, steps are not created equal. Some represent *Decision Forks*—pivotal junctures where the agent dictates a subsequent search strategy or significantly narrows the hypothesis space—while others are routine execution steps.

To identify these forks without human bias, we employed a model-based evaluation. For successful trajectories in the VideoMME benchmark, we provided an expert evaluator (Gemini-2.5-Pro) with the query, the full interaction trace, and the final outcome. The evaluator was prompted to identify the single "Top-1 Fork Step" ($k_{\text{fork}}$) based on the definition: *"A pivotal juncture in a multi-step reasoning process that dictates the subsequent logical trajectory... where the process diverges into multiple potential paths."*

### C.2. Quantitative Analysis: The Entropy Gap

We computed the mean token entropy for each turn $k$ in a trajectory $i$, denoted as $H_{i,k}$. We then calculated the difference between the entropy of the identified Fork Step ($H_{i,k_{\text{fork}}}$) and the average entropy of that entire trajectory ($\overline{H_i}$).

Figure 5 (a) presents the distribution of the entropy difference $\Delta H = H_{i,k_{\text{fork}}} - \overline{H_i}$.

- **Positive Shift (79.2%):** The vast majority of identified fork steps exhibit a positive entropy difference. This confirms that when the agent makes critical decisions (e.g., pivoting from scanning to verification), its policy distribution becomes flatter (higher uncertainty), reflecting the active weighing of potential reasoning paths.

- **Negative/Neutral (20.8%):** The minority of cases where the fork step entropy is lower or equal to the mean typically correspond to short-horizon trajectories or easy queries where the reasoning path is linear and deterministic.

### C.3. Case Study: Entropy at a Fork Step

To illustrate this phenomenon, we analyze a specific trajectory regarding the query: *"Which company is featured in the video but not mentioned in the audio?"* (Figure 5 (b)).

1. **Routine Scanning:** The agent performs standard scanning. The policy is confident in this information gathering, resulting in low entropy ($H \approx 0.397$).

2. **The Fork Step:** A distinct entropy spike ($H \approx 0.927$) occurs when the agent identifies that options A, B, and C are explicitly mentioned in the audio. This observation acts as a critical filter, ruling out these candidates. The spike reflects the agent's pivotal decision to switch modalities ('get_frames') to visually verify the presence of the remaining option (American Express), rather than concluding immediately.

3. **Resolution:** Subsequent verification resolves the ambiguity, returning the agent to a lower entropy state ($H \approx 0.790$).

This confirms that entropy spikes serve as a reliable signal for identifying high-value reasoning steps that warrant amplified reinforcement.

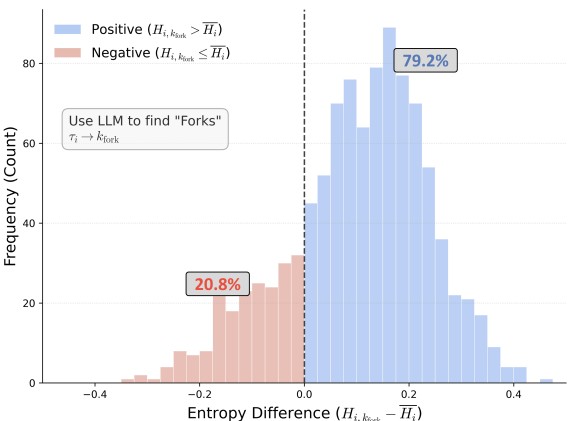

(a) Quantitative Analysis: Entropy Shift Distribution

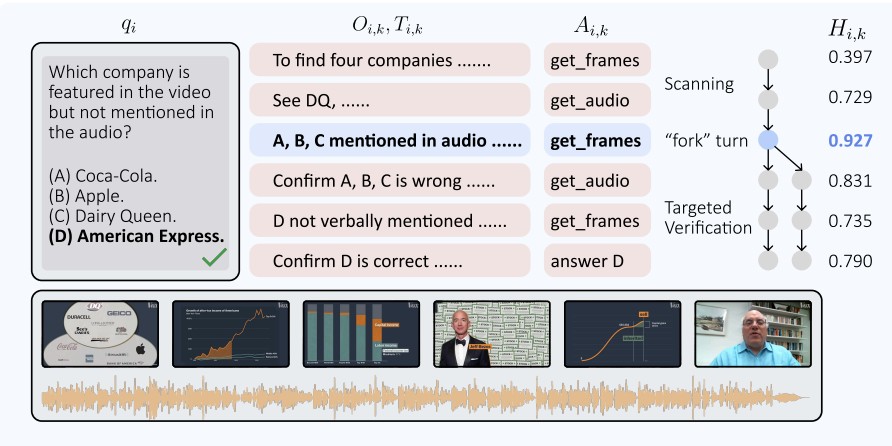

(b) Qualitative Case Study: Entropy Spike at Fork Step

*Figure 5.* **Correlation between Entropy and Critical Reasoning Steps. (a)** Histogram of the entropy difference $\Delta H = H_{i,k_{\text{fork}}} - \overline{H_i}$. **79.2%** of critical fork steps exhibit higher uncertainty than the trajectory mean. **(b)** A qualitative case study on the "Company" query. A distinct entropy spike (0.927) occurs at the "Fork Step" where the agent processes audio evidence (ruling out A, B, C) and decides to pivot to visual verification.

## C.4. Inference Runtime and Action Analysis

**Inference Runtime Analysis.** Table 7 reports measured wall-clock latency on a 100-sample LVBench subset. OmniAgent achieves lower wall-clock latency than Qwen2.5-VL-72B (66.8 s vs. 75.1 s) while reaching higher accuracy (51.0% vs. 47.0%). Compared to LongVT, an agentic baseline, OmniAgent is both more accurate and slightly faster. Qwen2.5-Omni-7B remains fastest in single-shot inference (34.8 s) but at substantially lower accuracy (41.0%). Notably, Qwen2.5-VL-72B requires $4\times$ A100 GPUs, whereas OmniAgent requires only 1.

*Table 7.* **Inference Runtime on LVBench (100 samples).** Wall-clock latency (seconds) and accuracy. OmniAgent achieves a favorable accuracy–latency tradeoff.

| Method | Frames | Model (s) | Wall-clock (s) | Acc. (%) |
|---|---|---|---|---|
| Qwen2.5-Omni-7B | 201 | 34.8 | 34.8 | 41.0 |
| Qwen2.5-VL-72B | 768 | 75.1 | 75.1 | 47.0 |
| LongVT | 793.8 | 64.1 | 67.6 | 42.0 |
| **OmniAgent (Ours)** | 201.6 | 56.0 | 66.8 | **51.0** |

# D. Qualitative Analysis

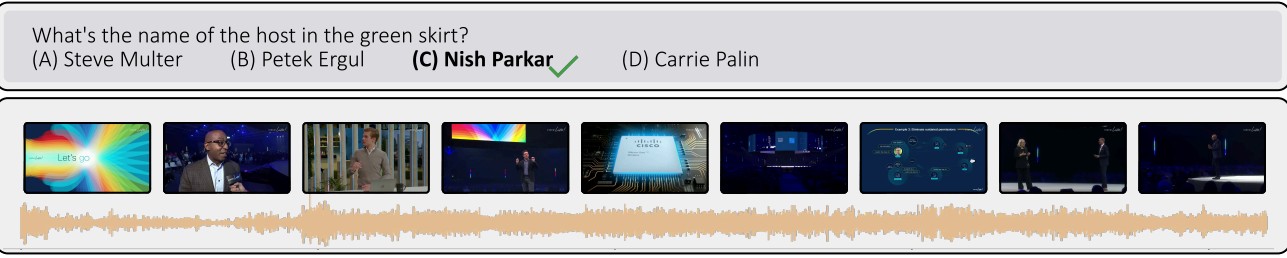

**What's the name of the host in the green skirt?**
(A) Steve Multer    (B) Petek Ergul    **(C) Nish Parkar** ✓    (D) Carrie Palin

**[Several OTAs omitted: OmniAgent performs broad temporal sampling over the full ∼140-minute video, then narrows around a suspected split-screen studio moment; the initial clips show no green skirt and no readable name tag.]**

**O:** I have identified a male host in a grey sweater (Studio A) and a male host in a grey suit (Studio B). I noticed a female host in a red jacket and another female host with curly hair. I missed checking the visual details of the first 200 seconds which might contain the studio intro. I need to locate a host wearing a green skirt. Potential segments are the intro, a roundtable discussion, or a different studio segment.
**T:** The studio segments are the most likely place for a host introduction. I will scan the first 200 seconds with higher density to identify the hosts and check their clothing. I will also listen to the audio in these segments to catch any introductions.
**A:** `get_frames(start=0.0, end=200.0, num=50)`

**O:** At 8.16s and 175.51s, a female host wearing a vibrant green dress/skirt is clearly visible standing next to two male hosts in the exhibition hall. She has dark hair. I need to verify her name by listening to the audio introduction or checking on-screen text if available.
**T:** I have located the host in the green skirt visually. Now I need to identify her name. The introduction usually occurs at the beginning of the clip. I will listen to the audio from 8.0s to 18.0s to hear if she is introduced.
**A:** `get_audio(start=8.0, end=18.0)`

**O:** Audio 8s-18s contained a teaser montage. I need the main intro audio to catch the name. Visually, the host in the green skirt was seen around 175s.
**T:** I will retrieve the audio segment immediately following the teaser (18s-48s) to catch the main introduction of the hosts. I am listening for a female voice introducing herself or being introduced.
**A:** `get_audio(start=18.0, end=48.0)`

**[Several OTAs omitted: OmniAgent triangulates around the host's repeated appearances (e.g., near 170s and around the split-screen at ∼424s), checks for lower-thirds, and then shifts to searching earlier intro-adjacent windows where name graphics commonly appear.]**

**O:** I need to confirm the identity of the female host in the green dress. I have not yet seen a text lower-third introduction in the intro sequence (0-30s). The host appears clearly at 6.21s and 14.48s. I will check the next segment of the video for a lower-third graphic that might have appeared after the intro montage.
**T:** I will scan frames from 48s to 160s to find any lower-third graphics identifying the hosts. This is a common place for an introduction to pop up.
**A:** `get_frames(start=48.0, end=160.0, num=30)`

**O:** At frame 55.72s, there is a clear lower-third graphic with the name "Nish Parkar" under the image of the female host in the green dress. She is wearing a green dress. She is identified as "Cisco Live Host #CiscoLiveEMEA". This matches option C.
**T:** I have found the definitive evidence. The lower third graphic at 55.72s identifies the female host in the green dress as Nish Parkar. I have also seen her earlier in the video (around 8s, 175s) in a green dress. This matches Option C. I will select Option C.
**A:** `answer(content="C")`

*Figure 6.* **Qualitative Example of Active Reasoning (MCQ).** OmniAgent first narrows the search space by revisiting the early broadcast window where host intros and lower-thirds typically occur, then visually confirms the green dress/skirt host at multiple timestamps. When audio checks yield only montage narration, it switches to a targeted scan of the intro-adjacent segment and locates the decisive lower-third text "Nish Parkar", grounding the final choice in on-screen evidence and selecting option **C**.

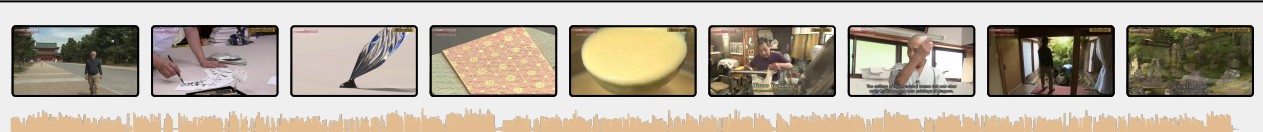

What is not true about Kyo-yuzen technique based on the video?    **(A) It contains 10 stages.** ✓
(B) The artworks contain very fine lines in some places.    (C) It includes the steaming process.    (D) Artists paint on the cloth

**O:** Initial scan initiated to locate segments discussing Kyo-yuzen technique.
**T:** The video is long (44 mins). I need to find sections discussing the Kyo-yuzen technique. I will perform a broad scan to identify scene changes and potential text overlays or speakers.
**A:** `get_frames(start=0, end=2645.51, num=60)`

**O:** Frames 403-582 cover the "Kyo-kanoko shibori" segment. I suspect "Kyo-yuzen" might be introduced in this context or nearby. Frame 762 shows a pattern that might be Kyo-yuzen. I need to verify the "10 stages" claim and the technique details.
**T:** I will inspect the audio in the shibori section to see if they mention "Kyo-yuzen" and its stages. If not found, I will scan the Kyo-yuzen section if it exists.
**A:** `get_audio(start=400, end=600)`

**O:** The audio confirms the segment at 400s-600s is about "Kyo-kanoko shibori", not Kyo-yuzen. They do mention "100,000 beads" (which relates to counting, not stages). I need to find the actual discussion on Kyo-yuzen to test the options.
**T:** I need to locate the specific section mentioning "Kyo-yuzen". The video seems to progress chronologically. Since 400s was Kyo-kanoko, I will scan forward to find the Kyo-yuzen segment, likely after a break. I will sample frames in the 600s-1200s range.
**A:** `get_frames(start=600, end=1200, num=50)`

**[Several OTAs omitted: OmniAgent systematically scans the video in large chunks (e.g. 1200–2400s, etc.) without finding the "Kyo-yuzen" segment.]**

**O:** I have thoroughly scanned the 1600s-1800s range and found it to be the "Sushi" segment. I have analyzed the "Kyo-kanoko" (400-600s) and "Kirikane" (750-1050s) segments. I have not found a dedicated "Kyo-yuzen" segment. To find the false statement about Kyo-yuzen, I must identify its features (e.g., is it painted on cloth or silk? does it involve 10 stages?). I will scan the 50s-150s range to see if Kyo-yuzen appears there, as I may have missed it in the initial sparse scan.
**T:** I need to locate Kyo-yuzen. I will check the 50s-150s range. I am looking for the specific Kyo-yuzen segment to evaluate the options.
**A:** `get_frames(start=50, end=150, num=50)`

**O:** In the museum segment (103s-123s), the text "Kyo-yuzen" is displayed. A panel details the process. It lists 10 stages: 5 Background dyeing, 6 Gold printing, 7 Steaming, 8 Rinsing, 9 Steam-smoothing, 10 Embroidery & gold leaf. Option A claims "It contains 10 stages", which matches the video. Option B claims "The artworks contain very fine lines in some places", which is true for the paintings. Option C claims "It includes the steaming process". The steaming is stage 7. Option D claims "Artists paint on the cloth". Kyo-yuzen involves "gilding" and "embroidery", not "painting on the cloth" in the traditional sense (painting with cloth). I suspect D is the false statement.
**T:** I will verify the audio to confirm the narrator's description of Kyo-yuzen's process. I will also check if "painting" is mentioned.
**A:** `get_audio(start=100, end=140)`

**O:** The audio at 100s-140s is decisive. The narrator states, "It's a very difficult and complex process involving more than 20 stages." He describes "In it, the painter paints directly onto the cloth", creating "very fine lines". He also says it involves "steaming" (stage 7).
Option A: It contains 10 stages. (FALSE. The audio explicitly says "more than 20 stages").
Option B: The artworks contain very fine lines in some places. (TRUE).
Option C: It includes the steaming process. (TRUE).
Option D: Artists paint on the cloth. (TRUE).
Since Option A is the only one that contradicts the video evidence ("more than 20 stages"), it is the correct answer for "What is not true".
**T:** I have confirmed via audio that the video says "more than 20 stages". My text observation of 10 stages was likely a miscount or reading of the board summary without full audio context. The audio "more than 20 stages" is the definitive false statement among the options.
**A:** `answer(content="A")`

*Figure 7.* **Qualitative Example of Audio-visual Collaborative Reasoning (MCQ).** OmniAgent performs targeted temporal queries to locate the *Kyo-yuzen* description, then cross-checks on-screen text with narration. It resolves the ambiguity about the number of stages by consulting the decisive audio evidence ("more than 20 stages"), selecting option **A** as the false statement while confirming the other options via narration (fine lines, steaming, and painting on cloth).

Answer with time ranges and do not output explanation. What are all the time ranges corresponding to the text query: "John Brassard was not happy about the meeting and adjourns the session by striking the gavel"? **GT: [1603, 1607]** **OmniAgent: [1603, 1607]**

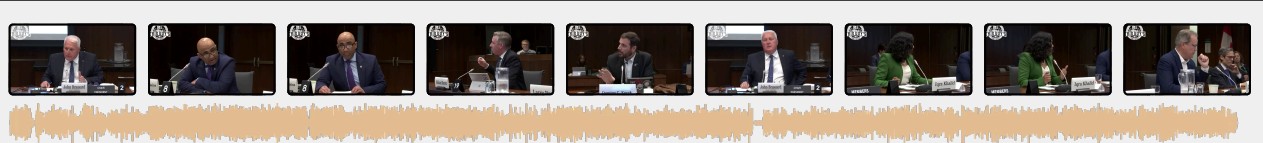

**O:** Initial scan initiated to locate the scene involving John Brassard and the gavel strike.
**T:** The video is long (26 minutes). I need to find the specific scene where a meeting is taking place and a gavel is used. I will start by scanning the entire video with 60 frames to identify the relevant segment.
**A:** `get_frames(start=0, end=1616.1, num=60)`

**O:** The video features John Brassard, a Chair/President, frequently. I see him gesturing and speaking in various segments. I suspect the event of him adjourning the session happens during a heated moment.
**T:** The prompt describes a specific action: "striking the gavel" because he is "not happy". I need to find the audio or visual of a gavel strike. Since I can't search for specific text, I will search for the audio cue of a gavel strike or the context of an altercation. I'll sample audio in the segments where he is speaking to identify the moment of conflict.
**A:** `get_audio(start=400.0, end=700.0)`

**[Several OTAs omitted: OmniAgent probes multiple short windows around ~700s with `get_frames`/`get_clip` to verify the suspected gavel cue, but fails to find an on-camera gavel strike.]**

**O:** Frame 756.32s clearly shows John Brassard, Chair (President), seated with the gavel visible on the desk. He appears calm, holding a pen. The clip ends at 760s. The previous clip (705-725s) showed him interacting with another speaker (Parm Bains) but no gavel strike. The audio cue at 700s might have been a false positive or a background sound. I need to find a moment where he is visibly agitated or gesturing emphatically, followed by a strike.
**T:** The 700s segment was a misinterpretation of the audio or a different scene. I need to search for another instance of Brassard speaking or being on camera. The 400-700 segment was a debate. I'll scan the later part of the video (1350-1600s) where he might have one last word or a different session ends. He is the Chair, so he calls adjournment.
**A:** `get_frames(start=1350.0, end=1616.0, num=30)`

**O:** Frames 1368-1588 show John Brassard gesturing and speaking agitatedly (pointing finger, touching head, hand on desk). This matches the "not happy" description. I need to find the specific moment he strikes the gavel to conclude the session.
**T:** The visual evidence from 1368s to 1588s confirms he is "not happy". The prompt states he "adjourns the session by striking the gavel". I haven't seen the strike yet. It likely occurs at the end of this rant, either around 1600s (before the video ends) or right after he stops speaking. I will examine the end of the video.
**A:** `get_clip(start=1600.0, end=1616.0)`

**O:** The clip from 1600s to 1616s captures the exact sequence: John Brassard declares "I am NOT happy with the way this went" (approx 1603s), immediately states "I am adjourning the meeting" (approx 1605s), and hits the gavel (approx 1606s). The clip ends with an outro logo.
**T:** I have located the exact moment. The sequence begins with his declaration of unhappiness and ends with the gavel strike. The timestamps align with the speech and the physical impact.
**A:** `answer(content="[[1603.0, 1607.0]]")`

*Figure 8.* **Qualitative Example of Temporal Grounding.** OmniAgent combines coarse full-video scanning with targeted audio checks to propose a candidate gavel event, actively falsifies the early false lead via localized probing, then pivots to a later visually-agitated segment and confirms the precise adjournment moment by querying the terminal clip. It outputs the exact interval covering "I am NOT happy" → "I am adjourning the meeting" → gavel strike: **[[1603.0, 1607.0]]**.

