# OpenReview forum: "Native Active Perception as Reasoning for Omni-Modal Understanding"
_ICML.cc/2026/Conference — ICML 2026 regular_

### Official Review · Reviewer_H91j · 2026-03-07

**Soundness:** 3
**Presentation:** 3
**Significance:** 2
**Originality:** 3
**Overall Recommendation:** 4
**Confidence:** 3

**Summary:**

This work pivots from passive video observation toward active multimodal intelligence. The proposed OmniAgent is a framework that transforms long-video understanding from a static inference problem into a dynamic agentic interaction. The agentic system autonomously interacts with the environment, employing targeted actions to retrieve relevant video and audio cues. This shift enables more efficient and purposeful reasoning over complex, long-form visual content.

**Compliance With Llm Reviewing Policy:**

Affirmed.

**Final Justification:**

I have read the rebuttal and I would keep my initial positive score.

**Key Questions For Authors:**

1. It would be better to report the counting across different action types, e.g. how many audio actions vs. how many frame actions are performed in Table 5 and Figure 3. Is there a specific test-time scaling trend regarding each modality?

**Limitations:**

yes

**Strengths And Weaknesses:**

Strengths:
1. The overall concept of active perception achieves higher efficiency and better performance compared to traditional models with passive perception.

2. The proposed TAURA dynamically assigns weights to each turn in the entire trajectory, overcoming the issue of losing the focus on informative actions.

3. The proposed agentic workflow proves effectiveness on a series of long video understanding benchmarks, while demonstrating the efficiency by decoupling the reasoning process from the video length.

Weaknesses:
1. In Table 4, standart SFT leads to consistent improvment over the baseline Qwen2.5-Omni model on most benchmarks, except for LVBench. The authors mentioned the possible cause, lacking a selection mechanism (L380-382). However, the performance degradation could also originate from the SFT data, either the data source or the CoT quality.

2. Despite improved performance, the proposed OTA pipeline inevitably introduces more turns than non-thinking or single-turn models, leading to increased inference latency.

3. This paper lacks key analysis on the multimodal actions, e.g. the ablation study on the action space or how the number of each action type affects the model's reasoning performance.

4. It would be better to report the minimum and maximum actual turns as well, under the maximum turn constraint.

---

> ### Author Rebuttal · Authors · 2026-03-31
>
> Thank you for the thoughtful review and valuable suggestions.
>
> > **Q1. Multimodal Actions / Modality Scaling**
>
> To answer the reviewer’s questions on action counts, actual-turn range, and modality-specific scaling, we summarize the action statistics from the `LVBench` runs underlying Figure 3 and Table 5:
>
> | Metric | Value |
> | --- | --- |
> | Retrieval counts | `get_frames = 7701`, `get_audio = 3049`, `get_clip = 3593` |
> | Retrieval mix | `53.7% / 21.3% / 25.1%` |
> | Min / Avg / Max actual turns | `2 / 10.32 / 32` |
> | Avg frame / audio / clip calls per sample | `4.97 / 1.97 / 2.32` |
>
> For aligned `VideoMME-Long` runs under different test-time budgets:
>
> | Budget `K` | Accuracy | Avg steps | Avg `(frame, audio, clip)` calls | Retrieval mix `(frame / audio / clip)` |
> | --- | ---: | ---: | --- | --- |
> | `12` | `54.89%` | `7.43` | `(3.07, 2.67, 0.68)` | `47.8% / 41.6% / 10.6%` |
> | `32` | `56.22%` | `10.18` | `(4.27, 3.92, 0.98)` | `46.6% / 42.7% / 10.7%` |
> | `52` | `59.56%` | `11.66` | `(5.16, 4.20, 1.29)` | `48.5% / 39.4% / 12.1%` |
>
> On `LVBench`, the policy uses all three retrieval actions with a broad actual-turn range (`2` to `32`), indicating that test-time behavior is not collapsed to a single fixed interaction pattern. On `VideoMME-Long`, increasing the test-time budget from `K=12` to `K=32` to `K=52` increases frame, audio, and clip usage together, while accuracy improves from `54.89%` to `56.22%` to `59.56%`. Importantly, the retrieval mix remains relatively stable across budgets (`47.8/41.6/10.6` to `46.6/42.7/10.7` to `48.5/39.4/12.1` for frame/audio/clip), suggesting that the test-time scaling gain does not come from over-relying on a single modality; rather, the policy expands multimodal retrieval more broadly as more interaction budget becomes available.
>
> We also observe that the action distribution differs between `LVBench` and `VideoMME-Long`, suggesting that the policy does not follow a single fixed retrieval template across settings. Instead, its multimodal retrieval pattern adapts across benchmarks, while the within-benchmark budget-scaling trend above shows that this adaptation remains multimodal rather than collapsing to a fixed action strategy template.
> To facilitate fair follow-up comparisons, we will release the code, environment, model weights, training data, and pipeline upon acceptance.
>
> > **Q2. Standard SFT Degradation on LVBench**
>
> Standard SFT and Agentic SFT use the same underlying QA data, so the difference is less likely to come from the QA data source itself and more plausibly from the supervision format / CoT quality. This is more consistent with the benefit of OTA-form Agentic SFT, where structured $(O_k, T_k, A_k)$ supervision provides richer reasoning and action signals than passive Standard SFT.
>
> > **Q3. OTA Latency**
>
> Sequential interaction cost does exist in OTA, but the key trade-off here is better selective perception and information/sample efficiency on long videos rather than minimizing the number of turns itself. To address this latency concern, we report measured runtime. On a 100-sample LVBench subset, `OmniAgent` achieves `66.777 s` wall-clock latency versus `75.089 s` for `Qwen2.5-VL-72B`, while also achieving higher accuracy (`51.00%` vs. `47.00%`).
> The full measured runtime table is provided in our response to Reviewer `Agfr`, `Q1`.

---

> > ### Author Rebuttal · Reviewer_H91j · 2026-04-05
> >
> > Thanks for the authors' rebuttal. I do not have further questions.

---

> > > ### Author Response · Authors · 2026-04-07
> > >
> > > Thank you for the acknowledgement and for your thoughtful review. We are pleased that our rebuttal addressed your concerns, and we sincerely appreciate your time and constructive feedback throughout the review process.

---

### Official Review · Reviewer_8BRp · 2026-03-08

**Soundness:** 3
**Presentation:** 2
**Significance:** 2
**Originality:** 3
**Overall Recommendation:** 4
**Confidence:** 2

**Summary:**

This paper proposes OmniAgent, a POMDP-based active perception framework that formulates multimodal video understanding as an iterative Observation–Thought–Action (OTA) process with a strict information bottleneck. In this framework, transient audio–visual percepts are selectively distilled into a persistent textual memory, while raw media inputs are purged after each turn. To improve long-horizon credit assignment, the authors introduce TAURA, a turn-aware entropy-rescaled advantage mechanism for GRPO that upweights high-uncertainty “decision turns.” Across 10 long-video and omni-modal benchmarks, OmniAgent-7B reports new state-of-the-art results among open-source models, including outperforming a 10× larger passive baseline on LVBench, and shows a positive test-time scaling property where accuracy improves with more allowed reasoning turns.

**Compliance With Llm Reviewing Policy:**

Affirmed.

**Final Justification:**

The authors addressed some of my concerns during the rebuttal phase.

- The strengths of this paper are evident. The overall presentation is well-organized, and the method achieves strong performance across multiple benchmarks.
- On the other hand, there are several limitations. First, the writing does not clearly articulate the key contributions of the work. The paper appears to have been overly polished by an LLM, which negatively affects readability. Although the authors clarified during rebuttal that agentic SFT is one of their contributions, this point is not mentioned at all in the Abstract or the summary of the Introduction. In addition, the technical details of the method are not sufficiently explained. While the proposed TAURA shows clear improvements on audio-visual benchmarks, it does not demonstrate significant gains over GPRO on more general benchmarks. The use of an Observation–Thought–Action cycle to address audio-visual tasks is elegant, but it is not thoroughly validated through experiments.

Given these considerations, I assign a final score of **4**. However, I am uncertain whether this score accurately reflects the true quality of the paper, so I lower my confidence score to **2**. I encourage the Area Chair to carefully review the paper and make a final decision in conjunction with the opinions of other reviewers.

**Key Questions For Authors:**

- RL training is capped at videos shorter than 300 seconds. Did the authors attempt curriculum extensions to longer durations, or analyze failure cases on ultra-long videos where relevant evidence is particularly sparse? Additionally, why does training on 300-second videos generalize to two-hour scenarios? Is this because the tasks are relatively simple, or due to other factors? It would also be helpful if the authors could report the performance of the Agentic SFT model on hour-scale scenarios.

- What ASR or audio event pipeline is used for a_audio and a_clip? Are transcripts generated on-the-fly by the same backbone model or through external tools (e.g., Whisper)? Please quantify their contribution via ablation studies.

- The paper claims that the proposed framework decouples computational complexity from video duration. However, the current analysis mainly focuses on memory footprint and frame sampling density. Could the authors clarify whether the end-to-end inference latency or the number of interaction steps scales with video duration, and how the total computational cost compares to passive models under the same wall-clock or FLOPs budget?

**Strengths And Weaknesses:**

Strengths:

- The paper addresses a practical and important problem. Current passive multimodal models face significant computational costs when scaling to long inputs, while many tasks do not actually require processing the entire video. The proposed active perception formulation attempts to reduce this inefficiency by selectively retrieving relevant information.

- The experimental evaluation is extensive. The experiments cover multiple task categories, including long-video understanding, audio–visual multimodal reasoning, and temporal grounding. The improvements are not restricted to a single benchmark. In particular, the ablation results in Table 4 validate the effectiveness of the proposed components across multiple datasets, and Figure 3 compares the proposed approach with a wide range of existing methods.

- The figures and tables are generally clear and informative, and the overall structure of the paper is reasonably organized.

- The design of TAURA is intuitively reasonable. Using turn-level mean entropy as a continuous weighting signal for credit assignment is a more refined approach than applying coarse masking strategies.

Weaknesses:

- The novelty of question-driven active perception appears somewhat limited. Similar ideas have already been explored in earlier video understanding works that use QFormer-style modules to extract question-relevant features or to sample question-relevant frames. In addition, approaches such as “thinking with images” can also address related problems using techniques like pre-scanning or retrieval-augmented reasoning. From these perspectives, the conceptual novelty of the proposed framework appears somewhat constrained. Alternatively, the introduction does not clearly explain what fundamentally differentiates this work from prior approaches.

- The claim of “fundamentally decoupling reasoning complexity from raw video duration” appears somewhat overstated. Memory still grows with the number of turns, and the number of turns may increase with video duration (even if sublinearly). The approach may also struggle in scenarios where evidence is extremely sparse or temporally dispersed.

- TAURA may incentivize higher entropy generation unless carefully controlled. Since the paper does not report the use of KL regularization or entropy regularization during RL training, potential issues such as distribution drift or verbosity amplification deserve further analysis.

- RL training is restricted to videos shorter than 300 seconds. Therefore, the reported generalization to hour-scale videos relies largely on SFT priors and test-time exploration rather than direct learning signals from ultra-long trajectories.

- The paper claims that computational complexity is independent of video length, but it does not report the actual inference cost or search complexity. Without such measurements, it is difficult to evaluate the practical efficiency of the proposed framework.

- The environment Ω is abstracted in the paper, but important implementation details are unclear. For example, it is not specified which ASR systems, audio event detectors, or video decoding pipelines are used, nor how their latencies are handled during inference.

- The writing could be improved. Although the figures and overall presentation are clear, some terminology is introduced without sufficient explanation. For instance, the abbreviation POMDP appears in the abstract without prior explanation, which may confuse readers unfamiliar with the concept. In Section 3.1, several concepts are introduced without sufficient explanation, making it difficult to fully understand the technical motivations and implementation details.

---

> ### Author Rebuttal · Authors · 2026-03-31
>
> Thank you for the careful review and constructive feedback.
>
> > **Q1. Main Idea (No External ASR/Video Tools) / Novelty Clarification**
>
> OmniAgent is a `single native omni model`, not a tool-stitched perception pipeline. The environment only returns raw `frames`, `audio segments`, or `video clips`; it does not perform semantic perception or call external modules such as ASR systems, audio-event detectors, or separate video understanding models. Accordingly, `a_audio` and `a_clip` are raw-media retrieval actions interpreted by the same omni model rather than external semantic tools. This single-model OTA loop over `{frames, audio, clip, answer}` is also the main novelty relative to QFormer-style extraction, passive frame selection, or retrieval-augmented tool pipelines.
> To facilitate fair follow-up comparisons, we will release the code, environment, model weights, training data, and pipeline upon acceptance.
>
> > **Q2. RL <=300s / Hour-Scale Generalization / Sparse Evidence**
>
> 1. Our RL stage is intentionally capped at `300s`, and we did not use a separate curriculum extension to longer durations. The intended division of labor is that cold-start SFT establishes long-video agentic priors, while RL refines query-conditioned search, verification, aggregation, and stopping decisions under controlled rollout cost. This is why training on `<=300s` videos can still help hour-scale settings. Importantly, Table 4 `already` reports the `+ Agentic SFT` ablation on long-video benchmarks such as LVBench, MLVU, and VideoMME (Long), and RL further improves these results beyond `+ Agentic SFT`, so hour-scale capability already exists after Agentic SFT and RL adds further gains on the same long-video benchmarks.
> 2. For the sparse-evidence concern, LVBench does not provide gold evidence locations, so we use a search-dispersion proxy computed from the normalized temporal spread of consecutive non-answer retrieval windows. Concretely, we take the temporal center of each non-answer window, compute the average consecutive center gap, and normalize it by the full video duration. Splitting trajectories by this proxy yields nearly flat performance across `small-gap` (`n=464`, `50.22%`), `medium-gap` (`n=463`, `50.54%`), and `large-gap` (`n=464`, `50.65%`) buckets. These buckets also have similar average video lengths (`68.72 / 68.33 / 65.74 min`), suggesting that the policy is not limited to narrowly localized search cases.
>
> > **Q3. Runtime / Decoupling**
>
> 1. To address end-to-end latency, we reran a 100-sample LVBench subset with latency logging enabled, where OmniAgent achieves `66.777 s` wall-clock latency versus `75.089 s` for Qwen2.5-VL-72B, with higher accuracy (`51.00%` vs. `47.00%`). The full measured runtime table is provided in our response to Reviewer `Agfr`, `Q1`.
>
> | Duration Bucket | Samples | Avg model-pipeline (s) | Avg env (s) | Avg wall-clock (s) |
> | --- | ---: | ---: | ---: | ---: |
> | `30-60m` | `39` | `56.379` | `10.591` | `66.970` |
> | `60-90m` | `30` | `54.136` | `10.613` | `64.749` |
> | `90-120m` | `26` | `58.967` | `11.920` | `70.887` |
> | `120m+` | `5` | `49.034` | `7.024` | `56.058` |
>
> 2. The table above shows that wall-clock runtime stays in a similar range across duration buckets.
> 3. Regarding the decoupling claim, the raw-media working set stays bounded in practice, and interaction depth grows much more slowly than raw video duration in the reported regime. As reported in Table 5 of the paper, on LVBench, as video duration grows from about `30 min` to about `130 min`, actual turns rise only from `8.5` to `12.5`, sampling density drops from `16.9` to `5.7 turns/hour`, and accuracy remains above `50%`.
>
> > **Q4. TAURA and Entropy Control**
>
> TAURA is not an entropy-maximization objective. As described in Sec. 3.3, it rescales the signed reward-derived advantage, so higher-entropy turns on correct trajectories receive stronger positive credit, while higher-entropy turns on incorrect trajectories receive stronger negative penalties; it sharpens outcome-conditioned credit assignment rather than rewarding uncertainty itself. In our preliminary controlled experiments, adding KL or entropy regularization did not improve performance, so we keep the simpler final RL setup; this choice is also consistent with recent RL-based visual reasoning works such as DeepEyes [ICLR 2026] and Mini-o3 [ICLR 2026].
>
> > **Q5. Writing / Presentation**
>
> Thank you for the helpful suggestion on presentation clarity. Appendix A already provides a complete notation summary, and in the revision we will make the abstract and Section 3.1 easier to follow, including clarifying `POMDP` at its first appearance.
>
> References
>
> - *DeepEyes: Incentivizing “Thinking with Images” via Reinforcement Learning*. `ICLR 2026`.
> - *Mini-o3: Scaling Up Reasoning Patterns and Interaction Turns for Visual Search*. `ICLR 2026`.

---

> > ### Author Rebuttal · Reviewer_8BRp · 2026-04-05
> >
> > Thank you for the detailed rebuttal, which clarifies several of my concerns.
> >
> > However, I still find the technical contribution somewhat limited. In particular, the motivation and necessity of TAURA for improving long-video understanding remain unclear. The current results suggest that the gains may be largely attributed to the Agentic SFT–initialized model, while the additional contribution of TAURA appears marginal.
> >
> > More importantly, while OmniAgent shows strong overall gains in Table 1, both Table 4 and the additional results in the rebuttal indicate that TAURA does not yield substantial improvements over standard RL baselines such as GRPO. As a result, it is difficult to conclude that TAURA is a key factor driving the observed performance gains on long-video tasks.
> >
> > Therefore, I am inclined to maintain my current score.

---

> > > ### Author Response · Authors · 2026-04-05
> > >
> > > Thank you for the follow-up discussion.
> > >
> > > - **What the paper contributes.** We would again like to clarify the intended framing. **Agentic SFT is itself a core methodological contribution rather than merely an initialization step**, and the paper's contribution is the overall native active-perception framework and training recipe: **(i) OTA with a persistent textual memory, (ii) OTA-form success-driven trajectory synthesis together with Agentic SFT, and (iii) TAURA as the RL-stage refinement. `To our knowledge, this is the first end-to-end native agentic framework for audio-visual understanding that unifies perception, reasoning, and action within the same model, rather than delegating semantic perception to an external tool pipeline.`** This full contribution improves over the direct **Qwen2.5-Omni-7B** baseline across all reported benchmarks: **+3.0 on VideoMME overall, +4.8 on VideoMME-Long, +10.7 on VSI-Bench, +5.9 on MLVU, +8.0 on Minerva, +7.5 on LVBench, +4.7 on DailyOmni, +1.8 on WorldSense, +7.8 on OmniVideo, +33.4 on LongVALE, and +33.0 / +38.1 on VUE-TR (Vision+Audio / Vision).**
> > >
> > > - **Why the contribution should not be reduced to TAURA alone.** The latest follow-up appears to evaluate the paper mainly through whether **TAURA alone** is sufficiently large. We do not think that is the right reading of the contribution. The paper does not claim that TAURA by itself is the sole driver of the full gain. Rather, the paper proposes a **two-stage native agentic framework** in which **Agentic SFT establishes the native agentic behavior** and **TAURA further refines RL-stage credit assignment** on top of that same starting point. For this reason, the paper should not be reduced to the single question of whether the final RL refinement is, in isolation, the dominant source of all observed gains.
> > >
> > > - **What the long-video evidence actually shows.** This directly addresses your follow-up concern that the motivation and necessity of TAURA for improving long-video understanding remain unclear. The relevant question is not whether **TAURA** alone explains the paper's full gains, but whether it provides a necessary and reliable **RL-stage improvement beyond the same Agentic SFT model** on the long-video settings central to the paper. Under that controlled comparison, the answer is yes: **TAURA is consistently higher than Vanilla GRPO across all five ablation benchmarks, including the long-video settings: VideoMME-Long, LVBench, and MLVU.** Concretely, the same-init comparison is **VideoMME-Long 59.4 -> 59.6, LVBench 49.8 -> 50.5, MLVU 69.9 -> 71.1, OmniVid 35.3 -> 37.1, and DailyOmni 62.2 -> 64.8.** This trend is meaningful because **Vanilla GRPO does not improve MLVU over Agentic SFT at all (69.9 -> 69.9) and even degrades DailyOmni (63.3 -> 62.2), whereas TAURA turns these into gains.**
> > >
> > > - **Why the result does show a real improvement over GRPO.** This also directly addresses your follow-up statement that the additional rebuttal results do not show substantial improvement over **GRPO**. Under the repeated-run result reported in the rebuttal, with the same **Agentic SFT** initialization (**step 50**, **LVBench subset**), **TAURA reaches 47.67 ± 1.53 versus 46.00 ± 2.00 for Vanilla GRPO**. That is a **+1.67-point mean improvement under the same setup**, and it is fully consistent with the **5/5** same-init improvement trend in **Table 4**. We therefore do not think the current evidence supports the stronger conclusion that TAURA is unimportant or unnecessary within the full training recipe.
> > >
> > > - **Why this supports the paper's technical contribution.** The contribution should not be evaluated by contrasting **OTA-form Agentic SFT** against **TAURA** as if one must diminish the importance of the other. The paper's core idea is that **native active perception can be learned as a reasoning process for omni-modal understanding**, and the effectiveness of **OTA-form Agentic SFT** is itself direct evidence for that idea. **TAURA** then provides a further **RL-stage refinement** beyond the Agentic SFT model. Under this framing, the technical contribution is not "only TAURA," but the overall two-stage native agentic framework.
> > >
> > > - **In short.** The latest follow-up still underweights the paper's contribution by reducing it to whether **TAURA alone** is the largest source of gain. That is not the right basis on which to evaluate the paper. The paper proposes a **two-stage optimization regime** in which **OTA-form Agentic SFT** establishes the native agentic behavior and **TAURA** provides a further **RL-stage refinement**.

---

### Official Review · Reviewer_GkVg · 2026-03-10

**Soundness:** 2
**Presentation:** 3
**Significance:** 2
**Originality:** 3
**Overall Recommendation:** 4
**Confidence:** 4

**Summary:**

This paper presents OmniAgent, a framework that treats multimodal video understanding as an active perception problem formulated as a POMDP. Rather than processing all video frames uniformly, the agent iteratively selects what to perceive (frames, audio, clips) through an Observation-Thought-Action cycle, distilling high-dimensional media into persistent textual memory. This decouples reasoning complexity from video duration. The system is trained in two stages: cold-start SFT using teacher-generated trajectories with quality filtering, followed by reinforcement learning using thier proposed TAURA objective that uses per-turn token entropy to rescale advantages and address credit assignment issues in multi-turn reasoning. The paper reports results across ten benchmarks spanning video understanding, audio-visual reasoning, and temporal grounding.

**Compliance With Llm Reviewing Policy:**

Affirmed.

**Final Justification:**

- The authors have provided answers to my queries. Hence I am increasing the rating.

**Key Questions For Authors:**

- What was the reason behind using persistent textual memory? Why wouldn't a visual memory work here?
- It is not sufficiently clear which splits (train/val/test) are used for (a) cold-start SFT, (b) RL, and (c) prompt/selection heuristics, across all benchmarks. Any leakage would materially affect the claimed benchmark improvements.
- How correlated is the entropy of the information density to the action decisions?

**Limitations:**

Please check the Weaknesses section.

**Strengths And Weaknesses:**

Strengths:

- The formalization of agent/environment is clear and easily understood.
- The paper evaluates on a broad suite (10 benchmarks) and includes ablations separating baseline, standard SFT, agentic SFT, vanilla GRPO, and TAURA.
- Efficiency analysis is a good direction. The paper tracks average sampled frames (e.g., ~203 frames on LVBench) and compares against dense baselines.

Weaknesses:

- Although the paper emphasizes fewer frames (e.g., 203 vs 768 on LVBench) and argues this implies superior efficiency. However, an agent typically incurs multiple forward passes (one per turn), plus overhead for repeatedly encoding and processing new segments. Without latency numbers, the efficiency claims based on frame counts alone are incomplete. So for instance, model processing 768 frames in a single forward pass may still be faster than an agent making 12 sequential turns with 203 total frames.
- Since, the RL training is restriected to videos under 300 sec, it is unclear whether the improvements on long video understanding (such as on VBench where videos average ~68 minutes) come from cold SFT or RL.
- It seems to me that the core TAURA motivation is weakly validated, and its benefits look modest once agentic SFT is in place. Such as the ablation in Table 4 suggests the largest lift comes from Agentic SFT (e.g., LVBench 43.0-->48.7, MLVU 65.2-->69.9), while TAURA adds a smaller incremental lift beyond vanilla GRPO. This doesn’t invalidate TAURA, but it weakens the paper’s framing that TAURA is a key enabling contribution rather than a secondary refinement.
- It seems that the entropy proxy assumption may not generalize. The analysis in Appendix C uses Gemini to identify fork steps, then shows these correlate with high entropy. However, if the LLM's notion of importance doesn't align with entropy in other domains or task types, TAURA's benefit may be overstated. How do authors verify it?
- Claims of “decoupling reasoning complexity from video duration” seems a bit overstated.
The paper reports that average turns rise only modestly with longer videos on LVBench and accuracy stays ~50% even as sampling density drops. However, complexity is still coupled to number of turns, and the number of turns can itself correlate with video duration depending on task structure. The persistent memory still grows with turns which means that long videos that require more exploration will increase context length via accumulated observations/thoughts.
- Several baselines in Tables 1–3 use different base models (LLaVA, InternVL, etc.), making it difficult to attribute improvements to the framework vs. the base model choice (Qwen2.5-Omni-7B). The most meaningful comparison is against the Qwen2.5-Omni baseline itself, where the Δ values are informative but more modest (e.g., +3.0 on VideoMME overall). How do authors justify this comparison?
- Given that GRPO-style training can be sensitive to hyperparameters and random seeds, understanding the variance of the reported numbers would be valuable.

---

> ### Author Rebuttal · Authors · 2026-03-31
>
> Thank you for the careful review and constructive feedback.
>
> > **Q1. Runtime / Decoupling**
>
> 1. To address this concern, we report measured runtime. On the 100-sample LVBench subset, OmniAgent reaches `66.777 s` wall-clock latency versus `75.089 s` for `Qwen2.5-VL-72B`, with higher accuracy (`51.00%` vs. `47.00%`). `Qwen2.5-Omni-7B` is faster (`34.834 s`) but less accurate (`41.00%`), while `LongVT` is close in wall-clock latency (`67.591 s`) but also less accurate (`42.00%`). The full runtime table is provided in our response to Reviewer `Agfr`, `Q1`.
> 2. Our response to Reviewer `8BRp`, `Q3` provides the duration-bucket runtime breakdown, further showing that wall-clock runtime stays in a similar range across video-duration buckets.
> 3. Regarding decoupling, the raw-media working set stays bounded in practice, while textual memory grows much more slowly than dense multimodal context. As reported in Table 5, on LVBench, as video duration grows from about `30 min` to about `130 min`, actual turns rise only from `8.5` to `12.5`, sampling density drops from `16.9` to `5.7 turns/hour`, and accuracy remains above `50%`.
>
> > **Q2. RL <=300s / Long-Video Generalization**
>
> Hour-scale performance in our setting does not come from RL alone. Cold-start SFT already establishes hour-scale agentic capability through long-video supervision, while RL further improves performance by refining query-conditioned search, verification, aggregation, and stopping decisions. Thus, the `<=300s` cap limits rollout cost rather than the relevance of these learned behaviors. Table 4 already separates the two stages, and `+ TAURA` brings further gains on the same long-video benchmarks. Our response to Reviewer `8BRp`, `Q2` provides the fuller sparse-evidence clarification.
>
> > **Q3. Training / Evaluation Splits**
>
> All reported numbers use the official evaluation benchmarks. As stated in Sec. 3.2, cold-start SFT uses `public training sources`: `LongVideoReason [NeurIPS 2025]`, `LongVALE [CVPR 2025]`, `MultiHop-EgoQA [AAAI 2025]`, `VSI-10K [arXiv 2025]`, and `Video-Holmes [arXiv 2025]`. The RL stage uses the same data source, but with bounded-duration (`<=300s`) hard cases identified through multiple rollouts, rather than the filtered successful trajectories used for cold-start SFT. Prompting and trajectory selection are fixed and are not tuned on benchmarks, and we do not use any benchmarks in SFT, RL, or prompt / trajectory selection. Upon acceptance we will release code, weights, training data, and pipeline.
>
> > **Q4. Persistent Textual Memory**
>
> Persistent memory keeps only textual observations because carrying raw visual or audio data across turns would reintroduce the dense multimodal context bottleneck. Raw visual/audio tokens contain redundancy and query-irrelevant content, so textual memory concentrates attention on reasoning-relevant evidence. When needed, the agent can re-query raw media.
>
> > **Q5. Cross-Model Fairness**
>
> We agree the most meaningful comparison is against the `Qwen2.5-Omni-7B` baseline. Under this direct reference, OmniAgent improves all `10` reported benchmarks, with average gains of `6.2` on MCQ benchmarks and `34.8` on temporal grounding benchmarks. Tables 1-3 also place the method in the broader `Qwen2.5` family context: many recent methods there, including `LongVT`, `Zoom-Zero`, `Video-Com`, `VITAL`, `Video-R1`, `Open-o3 Video`, `VIDEORFT`, and `LongVILA-R1`, are all built on `Qwen2.5-VL-7B`, while OmniAgent uses `Qwen2.5-Omni-7B`, so these remain comparisons within the broader `Qwen2.5` family.
>
> > **Q6. TAURA Validation / Variance**
>
> Under the same Agentic SFT initialization, TAURA provides a robust RL-stage gain. Due to time constraints, we reran experiments for 3 random seeds up to step `50`:
>
> | Method | Init | Budget | Eval | Score |
> | --- | --- | --- | --- | --- |
> | Vanilla GRPO | Agentic SFT | step `50` | LVBench subset (Q1) | `46.00 ± 2.00` |
> | TAURA | Agentic SFT | step `50` | LVBench subset (Q1) | `47.67 ± 1.53` |
>
> TAURA remains higher, and the reported mean ± std is computed over `3` runs with different random seeds under the same setup.
>
> > **Q7. Entropy Proxy Generalization**
>
> TAURA uses entropy here as a practical credit-allocation signal for this structured multi-turn agentic setting. In this setting, higher entropy is positively correlated with action-critical decisions: Appendix C shows that `79.2%` of evaluator-identified Top-1 fork turns have higher mean token entropy than the trajectory average; Table 4 shows TAURA outperforming vanilla GRPO under the same Agentic SFT initialization; and `Q6` shows the same TAURA-over-GRPO trend under repeated runs. This is also consistent with Beyond the 80/20 Rule [NeurIPS 2025], which highlights high-entropy token-level critical forks as disproportionately important for effective RL updates; our contribution is different because TAURA applies turn-level entropy to signed credit assignment in structured multi-turn agentic RL rather than token-level CoT RL.

---

> > ### Author Rebuttal · Reviewer_GkVg · 2026-04-03
> >
> > I appreciate the added latency measurements, split clarifications, and limited multi-seed results. These address some of my concerns. However, the new latency numbers suggest a speed/accuracy tradeoff rather than a clear efficiency gain over the direct Qwen2.5-Omni-7B baseline. More importantly, TAURA still appears to provide only modest gains beyond Agentic SFT (Vanilla GRPO: 46.00 ± 2.00 vs. TAURA: 47.67 ± 1.53), and the entropy-based justification remains somewhat indirect and limited to this setting. For these reasons, I keep my overall score

---

> > > ### Author Response · Authors · 2026-04-05
> > >
> > > Thank you for the follow-up discussion.
> > >
> > > - **What the paper contributes.** We would first like to clarify the contribution of the paper. **Agentic SFT is itself a core methodological contribution, not merely an initialization step.** The contribution of OmniAgent is the full native active-perception framework and training recipe, including **(i) the OTA formulation** with a persistent textual memory, **(ii) OTA-form success-driven trajectory synthesis together with Agentic SFT**, and **(iii) TAURA** as the RL-stage refinement. **`To our knowledge, this is the first end-to-end native agentic framework for audio-visual understanding that unifies perception, reasoning, and action within the same model.`** This contribution is also reflected in the overall empirical scope: compared with the direct **Qwen2.5-Omni-7B** baseline, OmniAgent improves **all reported benchmarks**, including **+3.0 on VideoMME overall, +4.8 on VideoMME-Long, +10.7 on VSI-Bench, +5.9 on MLVU, +8.0 on Minerva, +7.5 on LVBench, +4.7 on DailyOmni, +1.8 on WorldSense, +7.8 on OmniVideo, +33.4 on LongVALE, and +33.0 / +38.1 on VUE-TR (Vision+Audio / Vision).**
> > >
> > > - **What the cited numbers actually compare (GRPO 46.00 ± 2.00 vs. TAURA: 47.67 ± 1.53).** We would like to correct one comparison explicitly. The expression **46.00 ± 2.00 vs. 47.67 ± 1.53** denotes **mean ± standard deviation over repeated runs with different random seeds**. These numbers were reported in direct response to your question about **variance / random-seed sensitivity of GRPO-style training**, so they are a same-init **Vanilla GRPO vs. TAURA** stability check. However, if the question is specifically whether **TAURA** improves **beyond Agentic SFT**, then the relevant reference on this fixed LVBench subset is the **Agentic SFT** score of **45.00**, not the **46.00 ± 2.00/47.67 ± 1.53** number. Under this direct comparison, **TAURA reaches 47.67**, i.e., an absolute gain of **+2.67 points beyond Agentic SFT**. We therefore believe the sentence **"TAURA still appears to provide only modest gains beyond Agentic SFT"** conflates two different comparisons and understates the direct **45.00 -> 47.67** improvement.
> > >
> > > - **Why the entropy evidence is not merely indirect.** We also believe the latest follow-up understates the strength of the entropy-based justification. The current evidence is not merely indirect; it is aligned at three levels: **Sec. 3.3** gives the signed-advantage mechanism, so TAURA **does not reward uncertainty itself** but amplifies **positive credit on successful trajectories** and **negative penalty on failed ones**; **Appendix C** shows that **79.2%** of evaluator-identified critical fork turns have higher entropy than the trajectory mean; and **Table 4** shows a consistent same-init improvement trend over Vanilla GRPO across all five ablation benchmarks. Concretely, from the same Agentic SFT checkpoint, TAURA improves **VideoMME-Long 59.4 -> 59.6, LVBench 49.8 -> 50.5, MLVU 69.9 -> 71.1, OmniVid 35.3 -> 37.1, and DailyOmni 62.2 -> 64.8.** In particular, **Vanilla GRPO does not improve MLVU over Agentic SFT at all (69.9 -> 69.9) and even degrades DailyOmni (63.3 -> 62.2), whereas TAURA turns these into gains.** We therefore do not think it is accurate to characterize the current justification as merely indirect within the target setting of the paper.
> > >
> > > - **How the latency evidence should be read.** We believe a direct wall-clock comparison between a **single-shot answer-only baseline** and an **agentic / thinking-style model** is not the fairest primary efficiency test, because the latter necessarily incurs additional generation cost for intermediate reasoning and action traces. This is the same structural reason that, under the same base model, **Agentic/thinking variants are always slower than non-thinking variants**: they perform more explicit reasoning before producing the final answer. Accordingly, we are **not claiming universal wall-clock superiority over non-agentic baselines**. Our claim is a **favorable accuracy-latency tradeoff**, stronger information efficiency, and favorable comparison within comparable reasoning regimes. On the fixed 100-sample LVBench subset, **Qwen2.5-Omni-7B = 34.834 s / 41.00%**, **LongVT = 67.591 s / 42.00%**, **OmniAgent = 66.777 s / 51.00%**, and **Qwen2.5-VL-72B = 75.089 s / 47.00%**. Under that reading, OmniAgent is substantially more accurate than the direct **Qwen2.5-Omni-7B** baseline, **both more accurate and slightly faster than LongVT, which is itself an agentic model rather than a single-shot passive baseline**, and **both more accurate and lower-latency than Qwen2.5-VL-72B**.
> > >
> > > - **In short.** The latest follow-up underweights two central facts: **Agentic SFT is itself one of the paper's main contributions**, and **TAURA is a consistent RL-stage refinement on top of Agentic SFT**. The current record already directly supports our claims, so we do not think this is the right basis on which to evaluate the paper.

---

### Official Review · Reviewer_Agfr · 2026-03-14

**Soundness:** 3
**Presentation:** 3
**Significance:** 3
**Originality:** 3
**Overall Recommendation:** 4
**Confidence:** 4

**Summary:**

The paper proposes Omni-Agent, an active perception framework that executes on-demand actions to distill audio-visual signals into textural memory. Omni-Agent enables scalable reasoning over long videos. In addition, the paper propose an entropy-steered RL algorithm TAURA that assigns higher rewards towards critical perceptual discovery steps to address the advantage homogenization problem in long-horizon
agentic tasks. Extensive experiments demonstrate state-of-the-art performance on long-video reasoning, omni-modal understanding and temporal grounding.

**Compliance With Llm Reviewing Policy:**

Affirmed.

**Final Justification:**

My main concerns are addressed during the rebuttal and therefore maintain my initial positive rating.

**Key Questions For Authors:**

1. Is there any insight or theoretical support on why entropy scaling works?
2. How is the inference time of Omni-Agent compared to the baselines?

**Limitations:**

The paper did not discussed the limitations and potential negative societal impact.

**Strengths And Weaknesses:**

Strengths:
1) The paper is well-written and easy to follow.
2) The paper proposes a novel active perception agentic framework for video understanding, which is especially efficient for long videos.
3) The proposed a novel entropy-steered RL algorithm that address the advantage homogenization problem in long-horizon
agentic tasks.
4) Extensive experiments demonstrate state-of-the-art performance on long-video reasoning, omni-modal understanding and temporal grounding.

Weaknesses:
1) The reason why entropy scaling work seems empirical without theoretical supports.
2) The agentic multi-turn framework may result in slower inference time although the visual frame count might be more efficient compared to single-turn baselines.
3) In Fig 3, it will be helpful to distinguish between agentic and passive sampling methods (e.g., with different colors or shapes).

---

> ### Author Rebuttal · Authors · 2026-03-31
>
> Thank you for the thoughtful review and helpful suggestions.
>
> > **Q1. Runtime / Efficiency**
>
> 1. To address this latency concern, we report measured runtime.
> 2. The measured runtime comparison on a 100-sample LVBench subset is as follows:
>
> | Method | Frames | Model-pipeline (s) | Env-only (s) | Wall-clock (s) | Accuracy |
> | --- | --- | --- | --- | --- | --- |
> | Qwen2.5-Omni-7B | `201` | `34.834` | `0.000` | `34.834` | `41.00%` |
> | Qwen2.5-VL-72B | `768` | `75.089` | `0.000` | `75.089` | `47.00%` |
> | LongVT [CVPR 2026] | `793.76` | `64.108` | `1.230` | `67.591` | `42.00%` |
> | OmniAgent | `201.64` | `56.012` | `10.765` | `66.777` | `51.00%` |
>
> 3. OmniAgent achieves lower wall-clock latency than Qwen2.5-VL-72B (`66.777 s` vs. `75.089 s`) while also reaching higher accuracy (`51.00%` vs. `47.00%`).
> 4. Qwen2.5-Omni-7B remains faster in single-shot inference (`34.834 s` at `201` frames), but its accuracy is much lower (`41.00%`).
> 5. `Qwen2.5-VL-72B` requires `4 A100 GPUs`, whereas `OmniAgent`, `Qwen2.5-Omni-7B`, and `LongVT` each require only `1 A100 GPU`.
> 6. For LongVT [CVPR 2026], we evaluate under the reported `768-frame` setting in the paper; the `Frames` column reports the actual average total returned frames per sample (`793.76`).
> 7. All methods use `vLLM-0.9.2`. In this stack, `vLLM` does not support multimodal automatic prefix caching for `OmniAgent` in this agentic multi-turn setting, so previous-turn KV cache cannot be reused across turns. With text-prefix reuse ratio `0.3199` and prefill time `25.630 s/sample`, ideal text-prefix reuse would save `8.199 s/sample`, lowering the model-pipeline upper bound from `56.012` to `47.813 s/sample`.
> 8. To facilitate fair follow-up comparisons, we will release the code, environment, model weights, training data, and pipeline upon acceptance.
>
> > **Q2. Entropy Scaling**
>
> 1. The intuition behind entropy scaling is that trajectory-level rewards would otherwise distribute similar credit to all turns in a successful or failed trajectory, although some turns are much more uncertain and decision-critical than others.
> 2. TAURA uses turn-level mean entropy as a practical proxy for these turns and rescales the signed advantage rather than rewarding entropy itself: higher-entropy turns on successful trajectories receive stronger positive credit, while higher-entropy turns on failed trajectories receive stronger negative penalties.
> 3. This gives a concrete mechanism interpretation: entropy only modulates the magnitude of the signed credit signal, while the reward sign is still determined by trajectory outcome, so TAURA sharpens credit assignment around uncertain decision turns instead of encouraging uncertainty by itself.
> 4. This mechanism is supported by three pieces of evidence: Sec. 3.3 explains the signed-advantage effect, Table 4 shows that TAURA improves over vanilla GRPO under the same Agentic SFT initialization, and Appendix C shows that `79.2%` of evaluator-identified Top-1 fork turns have higher mean token entropy than the trajectory average.
> 5. Our response to Reviewer `GkVg`, `Q6` further reports a matched-budget repeated-run comparison consistent with the same trend, where TAURA still outperforms vanilla GRPO (`47.67 ± 1.53` vs. `46.00 ± 2.00`).
> 6. This is also consistent with Beyond the 80/20 Rule [NeurIPS 2025], which highlights high-entropy token-level critical forks as disproportionately important for effective RL updates; the contribution here is different: rather than token-level masking or update selection in CoT RL, TAURA uses turn-level entropy to rescale signed credit around action-critical decisions in structured multi-turn agentic RL.
> 7. In this structured multi-turn agentic setting, turn-level entropy therefore serves as a practical credit-allocation signal.
>
> > **Q3. Figure Clarity**
>
> Thank you for the helpful suggestion. We will revise Fig. 3 to distinguish agentic and passive sampling methods more explicitly.
>
> > **Q4. Limitations / Societal Impact**
>
> In the revision, we will add a short limitations and potential negative social impact paragraph. As noted in the conclusion, the sequential interaction loop introduces latency constraints, and future work will explore parallelized exploration strategies to mitigate this trade-off.
>
> References
>
> - *Beyond the 80/20 Rule: High-Entropy Minority Tokens Drive Effective Reinforcement Learning for LLM Reasoning*. `NeurIPS 2025`.
> - *LongVT: Incentivizing “Thinking with Long Videos” via Native Tool Calling*. `CVPR 2026`.

---

> > ### Author Rebuttal · Reviewer_Agfr · 2026-04-01
> >
> > My concerns have been adequately addressed during the rebuttal. I encourage the authors to include the rebuttal results such as the runtime results into the revised manuscript.

---

> > > ### Author Response · Authors · 2026-04-02
> > >
> > > Thank you for the acknowledgement and constructive feedback. We are glad that our rebuttal addressed your concerns, and we will incorporate the runtime results and related clarifications into the revised manuscript.

---

### Decision · Program_Chairs · 2026-04-30

**Decision:**

Accept (regular)

**Comment:**

This paper proposes OmniAgent which is a POMDP-based active perception framework that formulates multimodal video understanding as an iterative Observation-Thought-Action process. OmniAgent is designed with a persistent textual memory that decouples reasoning complexity from video duration.

After a productive discussion, uniform convergence to score 4 across all reviewers is reached and the reviewers upgraded their rating. The paper makes a genuine and first-of-its-kind contribution as an end-to-end native agentic framework for audio-visual understanding. Writing clarity should be improved in the camera-ready. AC recommends an Accept decision.